# Casein Kinases 2-dependent phosphorylation of the placental ligand VAR2CSA regulates *Plasmodium falciparum*-infected erythrocytes cytoadhesion

Dominique Dorin-Semblat[1], Jean-Philippe Semblat[1], Romain Hamelin[2], Anand Srivastava[3], Marilou Tetard[4], Graziella Matesic[5], Christian Doerig[6], Benoit Gamain[1]*

1 Sorbonne Université, CNRS, Inserm, Centre d'Immunologie et des Maladies Infectieuses, CIMI, Paris, France, 2 Proteomics Core Facility, Ecole Polytechnique Fédérale de Lausanne, Lausanne, Switzerland, 3 National Institute of Animal Biotechnology (NIAB), Hyderabad, Telangana, India, 4 Department of Pediatrics, Stanford University School of Medicine, Stanford, California, United States of America, 5 Université Paris Cité and Université des Antilles, INSERM, BIGR, Paris, France, 6 School of Health and Biomedical Science, RMIT University, Bundoora, Australia

* benoit.gamain@inserm.fr

**Data Availability Statement:** All data are found in the manuscript and Supporting information files.

## Abstract

Placental malaria is characterized by the massive accumulation and sequestration of infected erythrocytes in the placental intervillous blood spaces, causing severe birth outcomes. The variant surface antigen VAR2CSA is associated with *Plasmodium falciparum* sequestration in the placenta via its capacity to adhere to chondroitin sulfate A. We have previously shown that the extracellular region of VAR2CSA is phosphorylated on several residues and that the phosphorylation enhances the adhesive properties of CSA-binding infected erythrocytes. Here, we aimed to identify the kinases mediating this phosphorylation. We report that Human and *Plasmodium falciparum* Casein Kinase 2α are involved in the phosphorylation of the extracellular region of VAR2CSA. We notably show that both CK2α can phosphorylate the extracellular region of recombinant and immunoprecipitated VAR2CSA. Mass spectrometry analysis of recombinant VAR2CSA phosphorylated by recombinant Human and *P. falciparum* CK2α combined with site-directed mutagenesis led to the identification of residue S1068 in VAR2CSA, which is phosphorylated by both enzymes and is associated with CSA binding. Furthermore, using CRISPR/Cas9 we generated a parasite line in which phosphoresidue S1068 was changed to alanine. This mutation strongly impairs infected erythrocytes adhesion by abolishing VAR2CSA translocation to the surface of infected erythrocytes. We also report that two specific CK2 inhibitors reduce infected erythrocytes adhesion to CSA and decrease the phosphorylation of the recombinant extracellular region of VAR2CSA using either infected erythrocytes lysates as a source of kinases or recombinant Human and *P. falciparum* casein kinase 2.

Taken together, these results undoubtedly demonstrate that host and *P. falciparum* CK2α phosphorylate the extracellular region of VAR2CSA and that this post-translational

**Funding:** This work was supported by the Fondation pour la Recherche médicale (FRM) (EQU202203014741) and the Laboratoire d'Excellence ParaFrap (ANR-11-LABX-0024) to BG. The funders had no role in study design, data collection and analysis, decision to publish, or preparation of the manuscript.

modification is important for VAR2CSA trafficking and for infected erythrocytes adhesion to CSA.

## Author summary

*Plasmodium falciparum*-infected erythrocytes (IEs) isolated from pregnant women suffering from malaria express at their surface a specific variant of the *P. falciparum* Erythrocyte Membrane Protein (PfEMP1) family, namely VAR2CSA. VAR2CSA-expressing IEs interact in a Chondroitin Sulfate A (CSA) dependent manner with placental syncytiotrophoblasts leading to IEs sequestration in the placenta. We have previously shown that the extracellular domain of VAR2CSA is phosphorylated on several residues and that these modifications enhance the adhesive properties of CSA-binding IEs. Here we show that the red blood cell and the parasite Casein Kinase 2α are involved in the phosphorylation of the extracellular region of VAR2CSA. We notably found that these enzymes phosphorylate a specific residue and enhance VAR2CSA adhesion to CSA, while inhibiting these enzymes with specific inhibitors diminished IEs cytoadhesion to CSA. We also found that the mutation of this residue in the parasite genome impairs VAR2CSA translocation to the surface of infected erythrocytes and then their binding to CSA. Our study provides new insights into the molecular basis for placental malaria, but also identify Casein Kinase 2α as potential targets for intervention.

## Introduction

Malaria remains a serious global health and socio-economic burden, causing 247 million clinical cases and over 619,000 deaths per year [1]. The most virulent forms of malaria are caused by the parasitic protist *Plasmodium falciparum*. The severity of the disease is related to the capacity of *P. falciparum* infected erythrocytes (IEs) to adhere to a range of surface receptors expressed at the surface of host cells within the microvasculature, such as CD36, ICAM-1 and EPCR [2–4]. *P. falciparum* erythrocyte membrane protein-1 (PfEMP1) mediates IEs cytoadhesion to host cells and is displayed on protrusions of the IEs membrane called knobs [5,6]. The knob-associated histidine rich protein (KAHRP) has been shown to be crucial for the anchoring of *P. falciparum* erythrocyte membrane protein–1 (PfEMP1) [7]. PfEMP1, encoded by the multi-copy *var* gene family (~ 60 *var* genes per genome), is a variant antigen associated with immune evasion that relies on antigenic variation and monoallelic exclusion, whereby a single *var* gene is expressed by a given parasite at any given time [8,9]. Whereas the intracellular acidic terminal segment (ATS) of PfEMP1 is conserved and interacts with KAHRP, the extracellular region displays an N-terminal segment (NTS) followed by various numbers of highly polymorphic Duffy-binding-like (DBL) domains and cysteine-rich inter-domain region (CIDR), associated to antigenic variation [6,10]. Sequestration of IEs in the placenta is a hallmark of placental malaria (PM) and is associated with numerous complications such as maternal anaemia, premature delivery, stillbirth, low birth weight and increased perinatal and maternal mortality [11]. Chondroitin sulfate A (CSA) expressed on the syncytiotrophoblast layer and in the intervillous space of the placenta is the primary receptor for IEs adhesion and sequestration in the placenta [12,13]. This interaction is mediated by a single *var* gene, the PfEMP1 variant VAR2CSA, on the parasite's side [14–16].

We have previously shown that the extracellular domain of VAR2CSA is phosphorylated on several residues and that the phosphorylation enhances the adhesive properties of CSA-binding IEs [17]. Hora *et al.* have reported that phosphorylation of the intracellular ATS domain of PfEMP1 by Human Casein Kinase 2 (CK2) increases its affinity for KAHRP and thus plays a crucial role in IEs cytoadhesion [18]. CK2 is a highly conserved pleiotropic member of the protein kinase superfamily and is expressed in nearly every eukaryotic tissue and cellular compartment [19]. This kinase phosphorylates hundreds of substrates and is involved in several key cell processes such as proliferation, differentiation, apoptosis, DNA damage and repair, and, of note, in the context of the present study, cell adhesion [20,21]. In fact, analysis of phosphoproteomic datasets suggests that CK2 could be responsible for more than 10% of the Human phosphoproteome [22]. Protein kinase CK2 appears to exist in tetrameric complexes consisting of two catalytic subunits and two regulatory subunits. In many organisms, distinct isoenzymic forms of the catalytic subunit of CK2 have been identified. For example, in humans, two catalytic isoforms, designated CK2α and CK2α' and a dimer of regulatory subunits CK2β are found [23,24]. In line with the multiple functions of mammalian CK2, a central regulatory role for numerous cell processes such as *P. falciparum* invasion of red blood cells [25], chromatin assembly [26], intra-erythrocytic development [27] and more recently gametocytogenesis [28] has also been suggested for the *P. falciparum* orthologue (PfCK2). Kinome analysis of *P. falciparum* [29] identified one catalytic subunit PfCK2α and two distinct regulatory subunits, PfCK2β1 and PfCK2β2 [27]

In the specific context of PM, we aimed to determine whether Human and/or *Plasmodium* CK2 are involved in the phosphorylation of the extracellular region of VAR2CSA.

## Results

### CK2 inhibitors affect cytoadhesion but not VAR2CSA surface expression or trafficking

To assess if CK2 inhibitors affect IEs cytoadhesion, mature or ring stage NF54 IEs expressing VAR2CSA (NF54CSA) were treated for 1 hour or 16 hours respectively with 50μM of DMAT (2-Di Methylamini-4,5,6,7-tetrabromo-1 H-benzimidazole) or TBCA (Tetrabromocinamic acid) [18]. These two compounds are selective CK2 inhibitors [30,31,32] and have been used previously in *P. falciparum* cytoadhesion studies [18]. DMSO-treated IEs were used as a control. Using static binding assays on immobilized CSA, we showed that a one-hour treatment of mature stage IEs with DMAT drastically reduced IEs cytoadhesion to CSA (75% inhibition; p = 0.0001); a lower but still substantial inhibition was observed with TBCA (25% inhibition, p = 0.0007) (Fig 1A). When ring stages were treated for 16 hours, a strong reduction of mature stage IEs cytoadhesion (about 85% inhibition) was observed with both CK2α inhibitors (p = 0.0001) (Fig 1B).

Since reduction of cytoadhesion could be associated with a reduced level of VAR2CSA at the IEs surface, VAR2CSA surface expression was measured by flow cytometry using purified rabbit polyclonal anti-VAR2CSA antibodies. We found that VAR2CSA is expressed at similar levels on the surface of IEs treated with DMSO or CK2 inhibitors (S1 Fig).

### CK2 inhibitors impair phosphorylation of VAR2CSA by IEs total protein lysates

To assess the putative role of CK2 in VAR2CSA phosphorylation, late stages IEs total lysates were prepared as a source of both red blood cells (RBCs) and parasite kinases and were incubated with recombinant VAR2CSA extracellular region (rDBL1-6), in the presence of

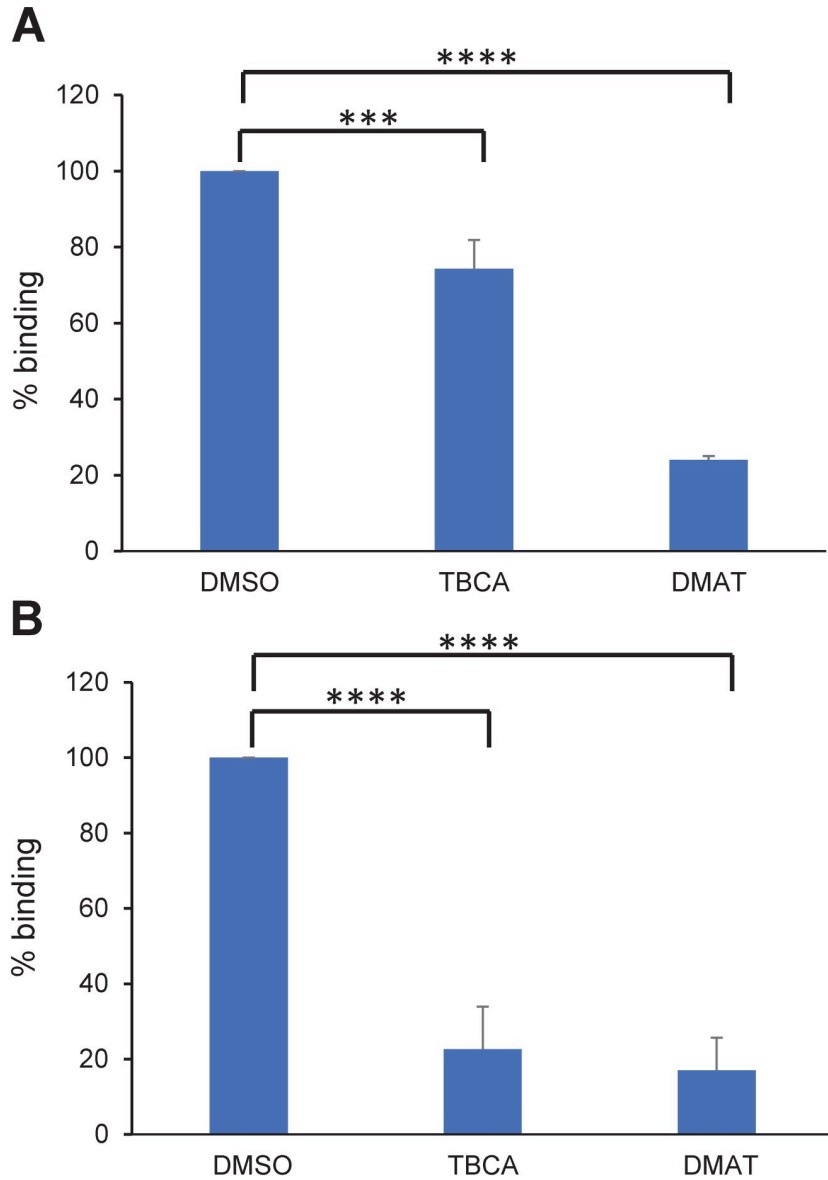

**Fig 1. CK2 inhibitors affect cytoadhesion but not VAR2CSA expression or trafficking. (A)** Parasitized RBCs at trophozoite stage and (**B**) ring stage were treated for 1 h or 16 h, respectively, with 50μM of TBCA or DMAT CK2 inhibitors prior to performing static binding assays on CSA and compared to the DMSO control. Bound mature stages IEs were counted in five random fields in 3 independent experiments. Results were expressed as a percentage of treated culture binding to CSA compared to DMSO-treated culture. Mean and Standard deviation are indicated. Statistics (Paired t test; ****p = 0.0001; ***p = 0.0007).

radiolabeled ATP and either DMAT, TBCA inhibitors or DMSO vehicle as a negative control. Equivalent amounts of rDBL1-6 were present in all assays as verified by Coomassie staining after SDS PAGE migration (Fig 2A and 2B **left panels**) prior to autoradiography exposure. The strong phosphorylation signal detected with IEs lysates incubated with DMSO was reduced in a dose-dependent manner by DMAT, and abolished at a 50μM concentration (Fig 2A **right panel**). Similar results were obtained with TBCA (Fig 2B **right panel**). Dose response was confirmed by quantification of the phosphorylation signal (S2 Fig). These results suggest

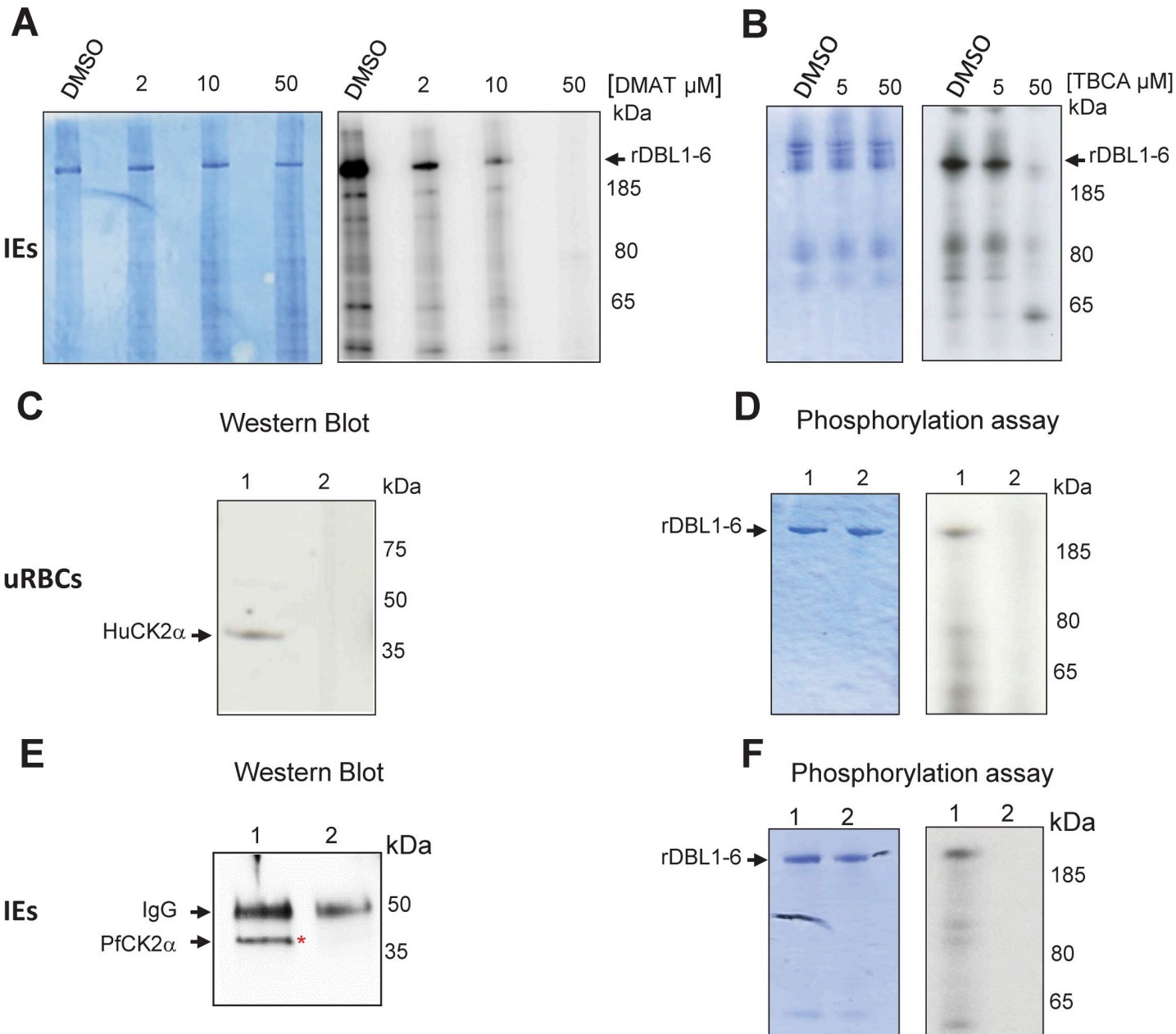

**Fig 2. Total IEs extracts, red blood cell HuCK2α and *P. falciparum* CK2α phosphorylate rVAR2CSA DBL1-6.** *In vitro* radioactive [γ—32P] ATP phosphorylation assays of recombinant His tagged VAR2CSA DBL1-6 protein (1µg in all assays) were performed in the presence of total IEs lysates and increasing concentrations of DMAT (**A**) or TBCA (**B**). (**A**) rDBL1-6 + DMSO; rDBL1-6 + DMAT 2µM; rDBL1-6 DMAT 10µM; rDBL1-6 + DMAT 50µM; (**B**) rDBL1-6 + DMSO; rDBL1-6 + TBCA 5µM; rDBL1-6 + TBCA 50µM. (**C**) Western blot analysis using a goat anti-Human CK2α; mouse anti-Human CK2α was used to immunoprecipitate the native HuCK2α from the membrane of uninfected erythrocytes. Control was performed with a non-specific mouse IgG isotype. Lane 1: immunoprecipitation with mouse anti-Human CK2α; lane 2: immunoprecipitation with mouse IgG control; (**D**) *In vitro* phosphorylation of rDBL1-6 by native HuCK2α. Immunoprecipitates were used in a standard [γ—32P] ATP phosphorylation kinase assay with 1µg of rDBL1-6. The reactions were loaded on a gel and stained with Coomassie (left panel) prior to being exposed for autoradiography (right panel). rDBL1-6 is indicated with an arrow. Lane 1: immunoprecipitation with mouse anti-HuCK2α; lane 2: immunoprecipitation with mouse control IgG isotype. (**E**) Western Blot anti PfCK2α**.** Rabbit anti-PfCK2α was used to immunoprecipitate the native kinase from total parasite lysates of IEs. Control was performed with the corresponding rabbit pre-immune immunopurified antibody. Samples were loaded on a SDS PAGE prior to transfer and detection with the same anti-PfCK2α. lane 1: immunoprecipitation with anti-PfCK2α; lane 2: immunoprecipitation with pre-immune immunopurified rabbit IgG. (**F**) *In vitro* phosphorylation of rDBL1-6 by endogenous PfCK2α. Immunoprecipitates were used in a standard [γ—32P] ATP phosphorylation kinase assay with 1µg of rDBL1-6. The reactions were loaded on a gel and stained with Coomassie (left panel) prior to being exposed for autoradiography (right panel). Lane 1: immunoprecipitation with immunopurified anti-PfCK2α; lane 2: immunoprecipitation with a pre-immune immunopurified antibody.

that CK2 from the red blood cell and/or *P. falciparum* are candidate kinases implicated in the phosphorylation of VAR2CSA.

### Red blood cell HuCK2α phosphorylates rVAR2CSA

To further investigate the involvement of Human CK2 in VAR2CSA phosphorylation, the native enzyme (HuCK2α) was immunoprecipitated from the membrane fraction of uninfected RBCs with a mouse anti-HuCK2α. Subsequent western blot analysis showed a band of ~ 40kDa, consistent with the molecular weight of HuCK2α (Fig 2C **lane 1**); no material was recovered when the immunoprecipitation was performed with a mouse IgG control antibody (Fig 2C **lane 2**). Immunoprecipitated HuCK2α was used in subsequent *in vitro* phosphorylation assay using rDBL1-6 as a substrate. A radiolabeled band corresponding to the size of rDBL1-6 is detected with the immunoprecipitated enzyme, indicating that HuCK2α can phosphorylate VAR2CSA extracellular region (Fig 2D **right panel lane 1**). No phosphorylation was observed with the mouse isotype control (Fig 2D **right panel lane 2**). The amount of protein loaded was verified by Coomassie staining (Fig 2D **left panel**).

### Endogenous PfCK2α phosphorylates rVAR2CSA

Since the IEs lysate used in Fig 2A and 2B contained both Human and the orthologous PfCK2α, we investigated the ability of native PfCK2α to phosphorylate rDBL1-6. Immunoprecipitation of the parasite kinase with an anti-PfCK2α, raised against a specific peptide, that do not cross-react with the Human CK2α (S3 Fig) was performed on total IEs lysates followed by western blot and kinase assays with rDBL1-6 as a substrate. Western blot analysis with anti-PfCK2α revealed a band corresponding to the size of PfCK2α (around 40kDa) in the PfCK2α immunopurified material (Fig 2E **lane 1**). This band was not detected in the immunoprecipitated material with the pre-immune antibody (Fig 2E **lane 2**). A radiolabeled band consistent with the size of rDBL1-6 was detected in the assay containing immunopurified PfCK2α (Fig 2F **right panel lane 1**). No phosphorylation of rVAR2CSA was observed using the immunoprecipitated material obtained with the pre-immune antibody (Fig 2F **right panel lane 2**). Equal loading was confirmed by Coomassie staining (Fig 2F **left panel**). To verify that the anti-PfCK2α antibody does not cross-react with the Human CK2α, we performed an immunoprecipitation with uninfected RBCs lysates. While the anti-Human CK2α immunoprecipitated the RBC kinase, no immunoprecipitation of the Human kinase was observed with the anti-PfCK2α. Furthermore, no cross-reactivity was observed by western blot with recombinant GST-PfCK2α (~65kDa) and MBP-HuCK2α (~78kDa). Taken together these results confirmed the specificity of both antibodies (S3 Fig).

### Native VAR2CSA is phosphorylated by rHuCK2α

To further demonstrate the involvement of Human CK2 kinases, we performed kinase assays using recombinant Human CK2α (rHuCK2α) and native VAR2CSA purified from IEs membranes. A rabbit polyclonal anti-NF54 VAR2CSA antibody was used to immunoprecipitate the protein from the NF54CSA (homologous) and FCR3CSA (heterologous) strains. Controls consisted of membrane fractions from uninfected red blood cells and FCR3 parasites selected for CD36 binding (and hence expressing different PfEMP1). Immunoconjugates obtained with the rabbit polyclonal anti-VAR2CSA were detected by western blot with a mouse monoclonal anti-VAR2CSA antibody targeting the ectodomain of the NF54 VAR2CSA variant. A band was observed in the immunoprecipitated material from the homologous NF54CSA IEs with a slightly higher molecular weight than the recombinant VAR2CSA rDBL1-6 positive control (Fig 3A **lane 3**), which is expected due to the presence of the transmembrane region

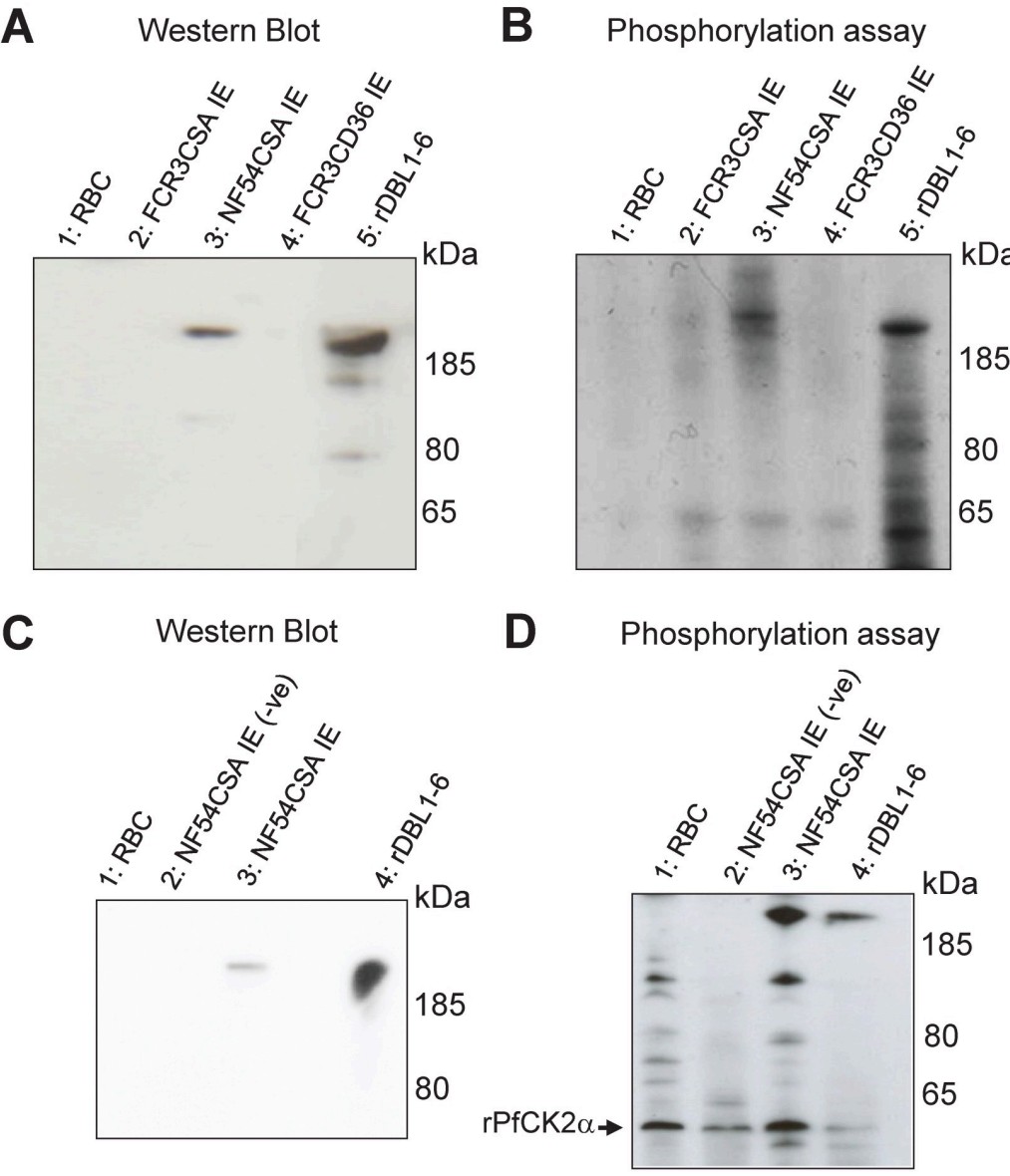

**Fig 3. Recombinant Human and *P. falciparum* CK2α phosphorylate immunoprecipitated VAR2CSA. (A)** A rabbit polyclonal anti-VAR2CSA was used to immunoprecipitate VAR2CSA from NF54CSA and FCR3CSA IEs membrane extracts. Membrane extracts from uninfected RBC and FCR3CD36 IEs were also processed using the same protocol. A fraction of immunoprecipitated material was loaded on a gel for western blot probed with a mouse monoclonal anti-VARCSA antibody. Lane 1: uRBC membrane extracts immunoprecipitation; lane 2: IEs FCR3CSA membrane extracts immunoprecipitation; lane 3: IEs NF54CSA membrane extracts immunoprecipitation; lane 4: IEs FCR3CD36 membrane extracts immunoprecipitation; lane 5: rDBL1-6 control. **(B)** *In vitro* phosphorylation of endogenous immunoprecipitated VAR2CSA by rHuCK2α. The immunoprecipitated materials were used as substrates in a standard [γ-32P] ATP phosphorylation kinase assay using 250ng of rHuCK2α. Phosphorylation of rDBL1-6 was performed as a control. Lane 1: immunoprecipitated uRBC membrane extracts; lane 2: immunoprecipitated FCR3CSA IEs membrane extracts; lane 3: immunoprecipitated NF54CSA IEs membrane extracts; lane 4: immunoprecipitated FCR3CD36 IEs membrane extracts; lane 5: rDBL1-6 control. **(C)** Immunoprecipitation with a rabbit polyclonal anti-VARCSA from uninfected and NF54CSA IEs membrane lysates. A rabbit IgG isotype was used as a negative control with NF54CSA IEs lysates. Western blot analysis of the immunoprecipitated material with a mouse monoclonal anti-VAR2CSA antibody; lane 1: uRBC membrane extracts immunoprecipitated material using anti-VAR2CSA polyclonal antibodies; lane 2: NF54CSA IEs lysates immunoprecipitated material with rabbit IgG isotype (-ve); lane 3: NF54CSA IEs lysates immunoprecipitated material using anti-VAR2CSA polyclonal antibodies; lane 4: rDBL1-6 control. **(D)** *In vitro* phosphorylation of endogenous immunoprecipitated VAR2CSA by rPfCK2α. The immunoprecipitated materials were used as substrates in a standard [γ-32P] ATP phosphorylation kinase assay using 250ng of rPfCK2α. Phosphorylation of rDBL1-6 was

performed as a control. lane 1: Immunoprecipitated uRBC membrane material; lane 2: Immunoprecipitated material with rabbit IgG isotype from NF54CSA IEs membrane extracts (-ve); lane3: Immunoprecipitated material with anti VAR2CSA antibody from NF54CSA IEs membrane extracts. Lane 4: rDBL1-6 control.

and cytoplasmic tail in the native protein, but not in the recombinant rDBL1-6 protein. No signal was observed in the RBC and FCR3CD36 (heterologous) IEs lysates. Immunoconjugates were then used in phosphorylation assays as substrates for rHuCK2α. A phosphorylated band corresponding to the size of the native full length VAR2CSA was detected by autoradiography in the immunoprecipitated material from NF54CSA IEs (Fig 3B **lane 3**). A weaker radiolabeled band was also observed with the immunoprecipitated material obtained from FCR3CSA IEs, likely due to cross-reactive epitopes between NF54 and FCR3 VAR2CSA (Fig 3B **lane 2**). Recombinant rDBL1-6 domain phosphorylated by rHuCK2α is shown as a positive control. No signal was observed in the RBC and FCR3CD36 IEs lysates (Fig 3B **lanes 1 and 4**). Taken together, these results indicate that the immunopurified VAR2CSA is phosphorylated by rHuCK2α. To verify that the immunoprecipitated VAR2CSA protein is not contaminated with PfCK2α or any other kinase activity, a phosphorylation assay was carried out after VAR2CSA immunoprecipitation from the membrane of IEs lysates (S4 Fig). No signal was observed in the control without addition of kinase **(lane 5)** while a radiolabeled band was detected after incubation of immunoprecipitated VAR2CSA with the rHuCK2α **(lane 3)**.

## Native VAR2CSA is also phosphorylated by rPfCK2α

We then assessed if native VAR2CSA is phosphorylated by recombinant PfCK2α (rPfCK2α).

Endogenous VAR2CSA expressed at the surface of NF54CSA-selected IEs was immunoprecipitated with the same polyclonal antibody as above and detected by western blot using the same mouse anti-VAR2CSA monoclonal antibody (Fig 3C **lane 3**). Immunoprecipitation performed on uninfected RBC membrane lysates or using rabbit IgG isotype control on NF54CSA IEs lysates did not yield any band. The recombinant rDBL1-6 that was loaded as a positive control (Fig 3C). The immunopurified materials were then used as substrates in rPfCK2α phosphorylation assays. A radiolabeled band corresponding to the size of native full length of VAR2CSA was detected in the material immunoprecipitated from NF54CSA IEs (Fig 3D **lane 3**). No phosphorylation was observed in control experiments, while the recombinant rDBL1-6 was phosphorylated by rPfCK2α (Fig 3D). Autophosphorylation of rPfCK2α was detected in all kinase reactions (indicated with an arrow). These results clearly indicate that native VAR2CSA is phosphorylated by rPfCK2α. To ensure that activity is associated with purified rPfCK2α rather than with co-purifying bacterial material, the catalytically inactive K72M mutant [27] was produced and purified under the same conditions as the wild-type kinase. Similar migration profiles of wild-type and K72M rPfCK2α were observed after SDS-PAGE and Coomassie staining (S5 Fig). While the wild-type enzyme could autophosphorylate and phosphorylate rDBL1-6, no phosphorylation was observed with the K72M mutant.

## *Plasmodium* and Human CK2 interact with rVAR2CSA *in vitro*

Having shown that Human and *Plasmodium* CK2α phosphorylate VAR2CSA, we wanted to confirm that VAR2CSA interacts with both casein kinases *in vitro*. rHuCK2α and rDBL1-6, expressed as MBP- and His-tagged proteins, respectively, were mixed and incubated for 30 min at 4°C prior to pull-down using amylose beads (Fig 4A and 4B **lane 2**) as described previously [27]. As a negative control, MBP alone was mixed with rDBL1-6 (Fig 4A and 4B **lane 1**).

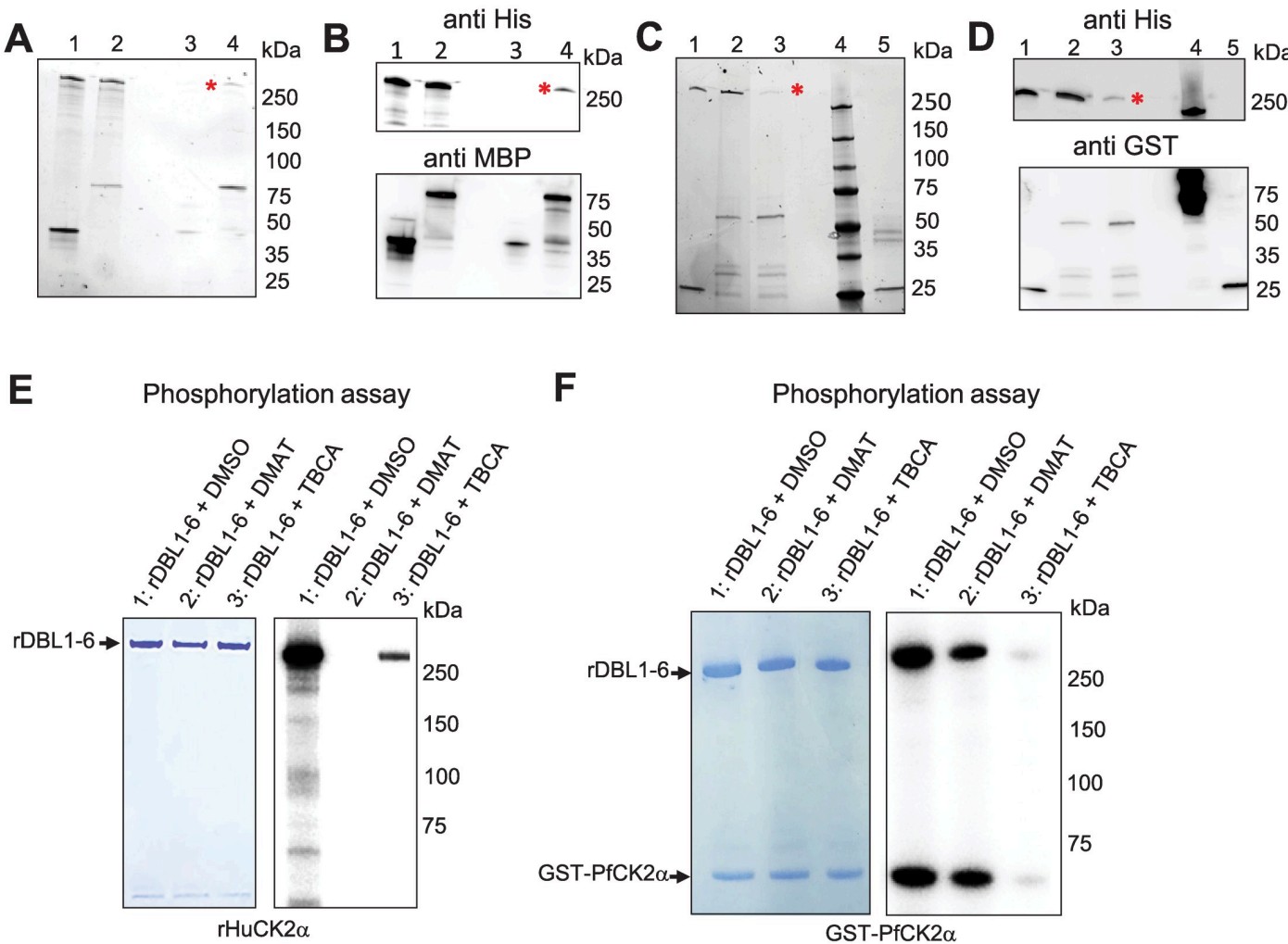

**Fig 4. Recombinant Human and *P. falciparum* CK2α interact and phosphorylate rDBL1-6. (A)** MBP-tagged HuCK2α or MBP alone were incubated with His-tagged rDBL1-6. Complexes containing the MBP-tagged proteins were then purified using amylose resin beads, and any bound His-tagged proteins were detected by stain-free SDS gel prior Western blot. **(B)** Western blot analysis using anti-His HRP **(upper panel)** or anti-MBP **(lower panel)**. Lane 1: MBP + His DBL1-6 (input); lane2: MBPHuCK2α + HisDBL1-6 (input); lane 3: MBP + HisDBL1-6 (bound fraction); lane 4: MBPHuCK2α + HisDBL1-6 (bound fraction). **(C)** GST-tagged PfCK2α or GST alone were incubated with HisDBL1-6. Complexes containing the GST-tagged proteins were then purified using glutathione agarose beads, and any bound His-tagged proteins were detected by stain-free SDS gel prior Western blot. **(D)** Western blot analysis using anti-His HRP (upper panel) or anti-GST (lower panel). Lane 1: GST + His DBL1-6 (input); lane2: GST-PfCK2α + HisDBL1-6 (input); lane 3: GST-PfCK2α + HisDBL1-6 (bound fraction); lane 4: MW: lane 5: GST + His DBL1-6 (bound fraction). **(E, F)** rDBL1-6 was used as a substrate for *in vitro* phosphorylation assays in the presence of [γ—32P] ATP with rHuCK2α **(E)** or rPfCK2α **(F)** in the presence or not of the CK2 inhibitors DMAT and TBCA at 50μM. Lane 1: rDBL1-6 + DMSO; lane2: rDBL1-6 + DMAT; lane 3: rDBL1-6 + TBCA.

To verify if the His-tagged VAR2CSA protein co-purifies with the MBP-tagged proteins, the pulled-down proteins were subjected to western blot analysis using an anti-His monoclonal antibody. The immunoblot confirms that DBL1-6 co-purified with MBP-HuCK2α but not with MBP alone. (Fig 4B **upper panel, lanes 3 and 4**). An anti-MBP western blot was performed to verify the presence and the amount of MBP-tagged proteins used in the assay (Fig 4B **lower panel**).

Similarly, using a GST-tagged PfCK2α recombinant protein and glutathione beads, we were able to co-purify rDBL1-6 with GST-PfCK2α but not with GST alone (Fig 4C and 4D) as shown using an anti-His antibody (Fig 4D **lanes 3 and 5 upper panel**). An anti-GST western

blot was performed to verify the presence and amount of GST-tagged proteins used in the interaction assay (Fig 4D **lower panel**).

## TBCA and DMAT inhibit *in vitro* rVAR2CSA phosphorylation by both recombinant kinases

Kinase assays were then carried out with rPfCK2α and rHuCK2α in the presence of the TBCA and DMAT CK2α inhibitors and VAR2CSA rDBL1-6 as a substrate (Fig 4E and 4F). Both inhibitors reduced the activity of both kinases, with 50μM DMAT displaying a stronger inhibitory effect on HuCK2α (Fig 4E) than on rPfCK2α (Fig 4F). 50 μM TBCA abolished VAR2CSA phosphorylation by rPfCK2α and highly reduced phosphorylation by rHuCK2α. These molecules inhibit both enzymes in a dose-dependent manner (S6 Fig).

## Identification of targeted VAR2CSA domains and phosphosites

Having demonstrated that both HuCK2α and PfCK2α phosphorylate VAR2CSA, we then aimed to identify which extracellular domains of VAR2CSA are phosphorylated. Phosphorylation assays were therefore carried out on single or multi-domain VAR2CSA recombinant proteins.

The single VAR2CSA domains DBL1X, DBL2X, DBL3X, CIDR, the multi-domains INT1CIDR, DBL1X-2X, DBL1X-3 X, DBL4ε-6ε and the full-length DBL1X-6ε were used as substrates in kinase reactions. All domains were produced in HEK293 cells except DBL2X, DBL3X, INT(BIS) CIDR and CIDR, produced in *E. coli* (Fig 5A and 5B **upper panel**). CIDR-containing proteins were phosphorylated by rHuCK2α, whereas no signal was observed for DBL1X, DBL2X and DBL3X domains (Fig 5A **lower panel**). In contrast, rPfCK2α caused a slight phosphorylation of the DBL1X and DBL3X and a stronger phosphorylation of DBL2X and CIDR domain (Fig 5B **lower panel**). The multi-domains INT1-CIDR, DBL1X-DBL2X and DBL1X-3X were phosphorylated by rPfCK2α, whereas no signal was detected with the multidomain DBL4ε-6ε corresponding to the C-terminal of the VAR2CSA protein (Fig 5B **lower panel**).

To determine the residues phosphorylated by rPfCK2α and rHuCK2α, mass spectrometry analysis was performed on rDBL1-6 incubated with either enzyme in the presence of cold ATP. Kinase reactions were loaded on an SDS-PAGE, and the rDBL1-6 band was excised for protein content analysis by liquid chromatography-tandem mass spectrometry (LC-MS/MS). This allowed us to identify several phosphosites (Table 1). Upon incubation of rDBL1-6 with rHuCK2α, only one phosphoresidue, S1068, was identified in the CIDR domain (Fig 5C). Interestingly, this phosphosite has an acidic environment and is predicted to be a CK2 target by the online prediction tool Net Phos3.1 [17].

While HuCK2α phosphorylated only S1068, five residues located in the N-terminal region of rDBL1-6 were phosphorylated by PfCK2α: S433 and S453 in INT1, T821 in DBL2X, S980 and S1068 in CIDR (Fig 5C). The highly conserved S433, which we found in our previous study [17] to be important for *in vitro* CSA binding, has been predicted as a CK2α phosphosite by NetPhos [17]. The identified phosphosites correlate with the radiolabeled phosphorylation mapping of single and multidomain VAR2CSA (Fig 5A and 5B).

## Effect of mutated phosphosites on phosphorylation of rVAR2CSA DBL1-6

To validate the phosphoproteomic dataset, the CIDR produced in bacteria, DBL1X-3X, and DBL1X-6ε proteins with S1068A mutation produced in HEK 293 cells were assayed in *in vitro* phosphorylation reactions with HuCK2α. Alanine substitution at Serine 1068 resulted in a substantial reduction of phosphorylation for all substrates (Fig 6A). To ensure that S1068 is

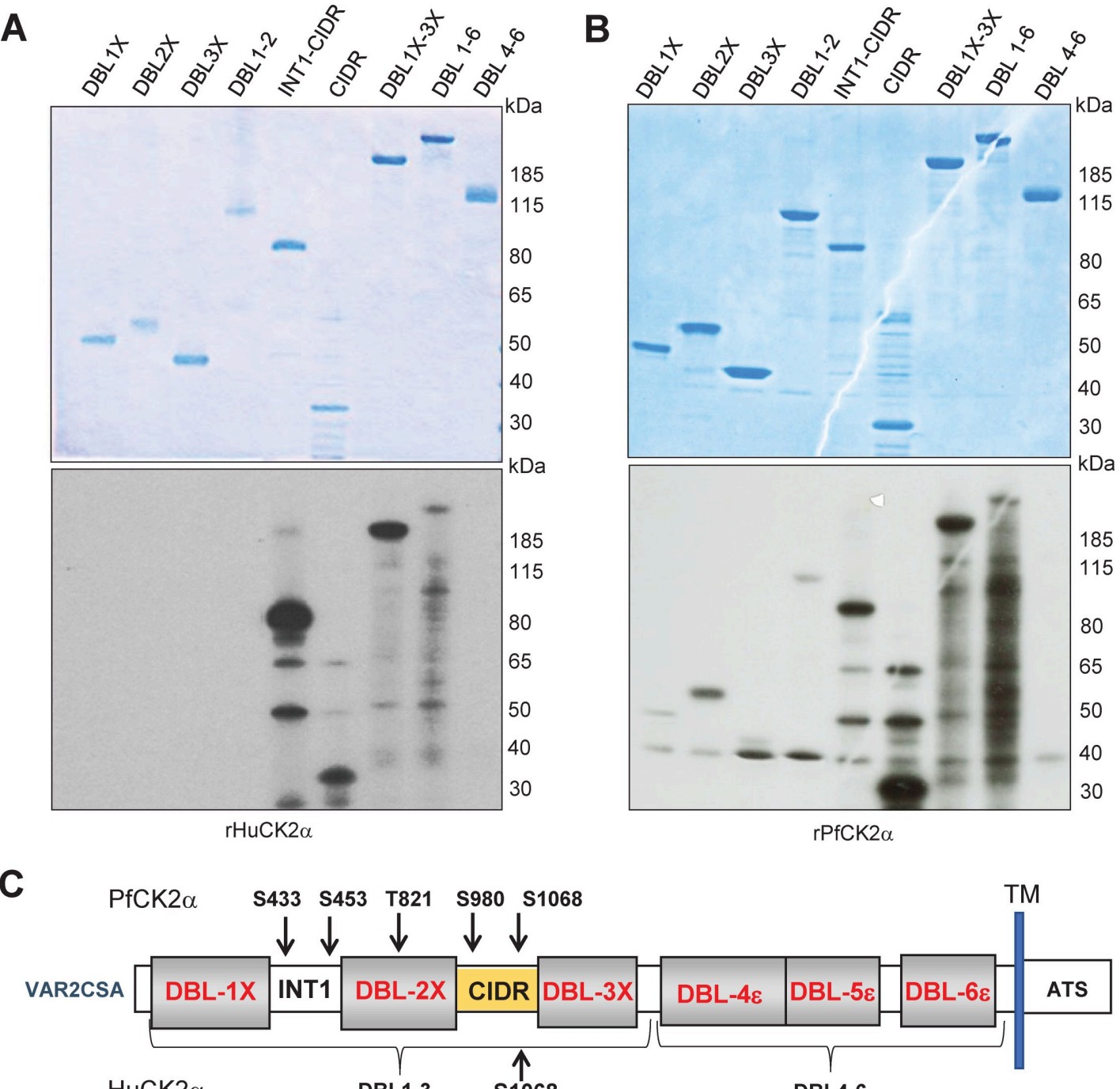

**Fig 5. Identification of targeted VAR2CSA domains and phosphosites. (A, B)** Mapping of phosphorylated VAR2CSA extracellular domains. Single and multidomains of recombinant VAR2CSA DBL1-6 were used as substrates for *in vitro* phoshorylation assays with rHuCK2α and r*Pf*CK2α. Autoradiograms are shown on the lower panels and the corresponding Coomassie-stained gels are in the top panel. **(A)** Phosphorylation assays in the presence of [γ—32P] ATP and rHuCK2α. **(B)** Phosphorylation assays in the presence of [γ—32P] ATP and r*Pf*CK2α. Lane 1: DBL1X; lane 2: DBL2X; lane 3: DBL3X; lane 4: DBL1X-DBL2X; lane 5: INT1-CIDR; lane 6: CIDR; lane 7: DBL1X-3X; lane 8: DBL1X-6ε; lane 9: DBL4ε-6ε. rDBL1-6 were used as substrates for *in vitro* cold phosphorylation assays with rHuCK2α and r*Pf*CK2α. **(C)** Schematic view of rDBL1-6 identified phosphorylation sites by r*Pf*CK2α and by rHuCK2α; TM: transmembrane domain; ATS: Acidic Terminal Segment corresponds to the cytoplasmic region of PfEMP1s proteins.

**Table 1. Summary of phosphorylation sites of rDBL1-6 after kinase assay with rHuCK2α and rPfCK2α.**

| Sequence | SEQUEST XCorr | Mascot Ion score | X! Tandem | Modifications | Observed | Charge | Delta PPM | Start | Stop | Rec Phosphosite | Native phosphosite |
|---|---|---|---|---|---|---|---|---|---|---|---|
| **HuCK2α** | | | | | | | | | | | |
| VSDEAAQPKFSDNER | 3,6 | 40,1 | 2,3 | Phospho (+80) | 591,5874 | 3 | - 0,6 | 1000 | 1014 | S1010 | S1068 |
| **PfCK2α** | | | | | | | | | | | |
| NNDEVcNcNESGIA**S**VEQEQISDPSSNK | 5,3 | 30,4 | 3,9 | Carbamidomethyl (+57), Carbamidomethyl (+57), Phospho(+80) | 1068,7643 | 3 | 1,6 | 361 | 388 | S375 | S433 |
| AcITHS**S**IK | 3 | 18,4 | 3,9 | Carbamidomethyl (+57), Phospho(+80) | 548,7465 | 2 | -0,1 | 389 | 397 | S395 | S453 |
| GGDGTAGSSWVK | 3,2 | 4,1 | | Phospho (+80) | 601,2483 | 2 | 0,5 | 759 | 770 | T763 | T821 |
| EYmNQWSCG**S**AR | 3,3 | 4,6 | | Oxidation (+16), Carbamidomethyl (+57), Phospho(+80) | 792,7866 | 2 | 3,3 | 913 | 924 | S922 | S980 |
| VSDEAAQPKF**S**DNER | 3,7 | 20,6 | | Phospho (+80) | 886,879 | 2 | 1,1 | 1000 | 1014 | S1010 | S1068 |

Identified phosphopeptides and their sequences are reported with their localization score.

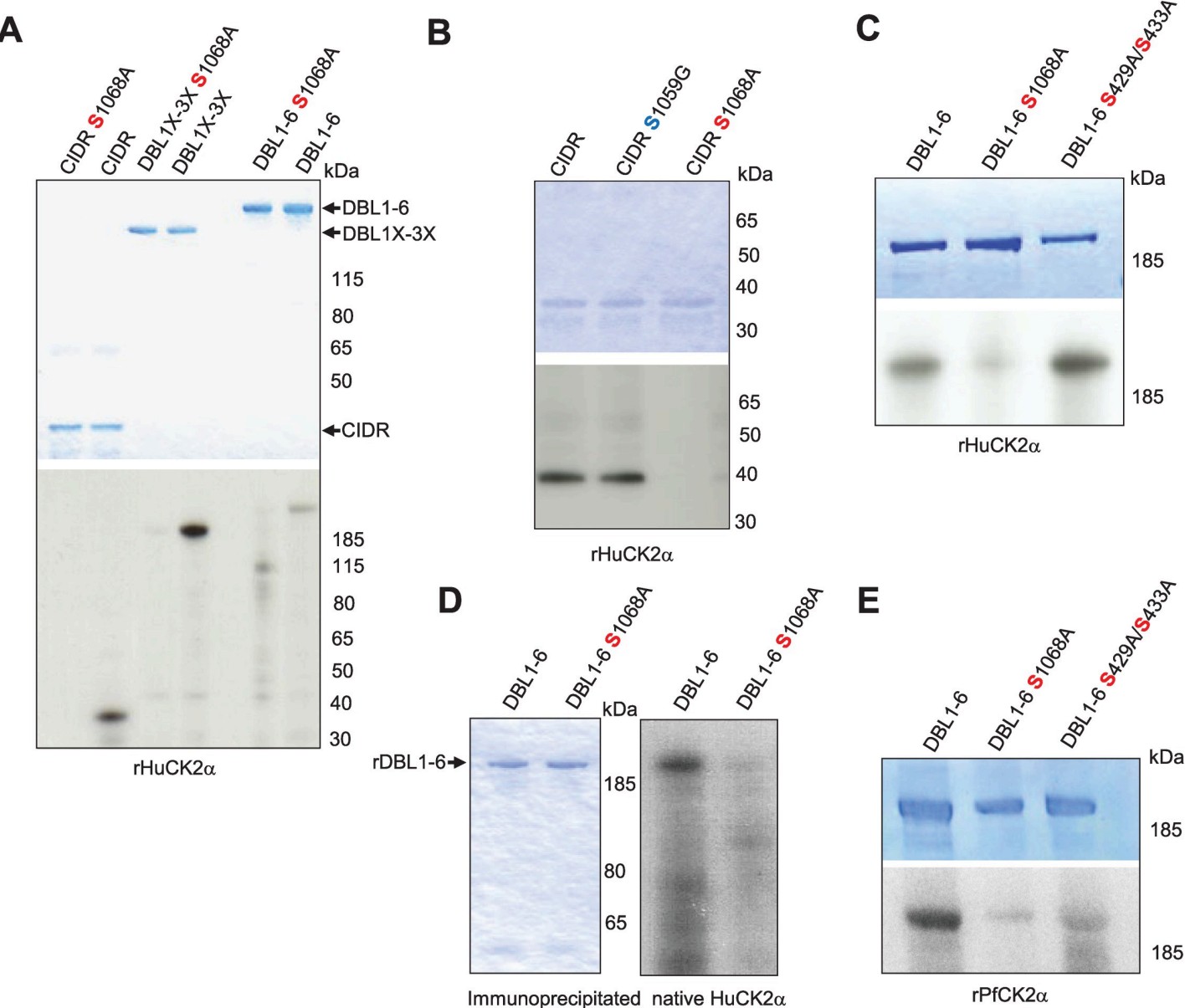

**Fig 6. Validation and effect of phosphosites mutation on VAR2CSA phosphorylation and CSA adhesion. (A, B, C).** The wild type and mutated VAR2CSA recombinant CIDR, DBL1X-3X and DBL1X-6 proteins indicated in the figure were tested for phosphorylation in a standard *in vitro* phosphorylation assays in the presence of [γ-32P] ATP with HuCK2α. **(D)** Wild type and S1068A DBL1X-6 recombinant proteins were tested for phosphorylation in standard *in vitro* phosphorylation assays in the presence of [γ-32P] ATP with endogenous immunoprecipitated HuCK2α. **(E)** The wild type and mutated rDBL1X-6, DBL1X-6 S1068A and DBL1X-6 S429A/S433A proteins indicated in the figure were tested for phosphorylation in a standard *in vitro* phosphorylation assays in the presence of [γ-32P] ATP with the wild type recombinant *Pf*CK2α or the catalytically inactive mutant K72M recombinant *Pf*CK2α. In all panels, similar amounts of wild-type and mutated proteins were loaded on SDS-PAGE, as shown by Coomassie blue staining.

the dominant phosphorylation site for HuCK2α, another point mutation, S1059G, was generated in CIDR protein, and kinase reactions were carried out. The signal was similar to that obtained with wild-type CIDR, confirming that HuCK2α does not phosphorylate this residue (Fig 6B). We also assayed the S429A/S433A double mutant generated in our previous study [17]. Similar signals were observed with rDBL1-6 WT and rDBL1-6 S429A/S433A proteins,

ruling out phosphorylation of these residues by HuCK2α, whereas almost no phosphorylation was detected with the VAR2CSA S1068A mutant, consistent with our phosphoproteomic data (Fig 6C).

Moreover, immunoprecipitated native HuCK2α from the membrane fraction of uninfected red blood cells failed to phosphorylate rDBL1-6 S1068A, whereas an efficient phosphate transfer occurs with wild-type rDBL1-6, further supporting the proposition that S1068 is the dominant phosphorylation site for native HuCK2α as well (Fig 6D).

The same mutants were used with recombinant PfCK2α. A reduced but not abolished signal was observed with both mutants (S1068A and S429A/S433A) compared to the wild-type VAR2CSA signal (Fig 6E), indicating that PfCK2α-mediated VAR2CSA phosphorylation occurs on several amino acids, including S1068 and S433, again consistent with mass spectrometry data.

### rDBL1-6 S1068A mutation impairs CSA-binding while its phosphorylation by rHuCK2α enhances CSA binding

Next, we evaluated the phenotypic effects of S1068 mutation on rDBL1-6 in *in vitro* binding assays to CSA and decorin, a glycoprotein consisting of a core protein and chains of CSA. An ELISA was performed with increasing concentrations of recombinant rDBL1-6 WT or rDBL1-6 S1068A. A slight, but significant, decrease (P = 0.0001) in adhesion was observed for the mutant protein on both CSA and decorin (Fig 7A), raising the possibility that S1068 contributes to adhesion. Since we have shown by mass spectrometry that the HuCK2α phosphorylates the S1068 residue and that the phosphorylation could contribute to VAR2CSA binding to CSA, we assessed the binding of rDBL1-6 to CSA at 4 different concentrations after addition or not of Human CK2a in the presence of ATP. Interestingly, we observed in 3 independent experiments a significant increase in the binding to CSA (p = 0.018; 0.03; 0.0008 and 0.0002 for respectively 0.2, 0.5, 1 and 2μg/ml rDBL1-6) when rVAR2CSA was preincubated with the kinase plus ATP compared to the condition without kinase, indicating that HuCK2α mediated VAR2CSA phosphorylation enhances binding to CSA (Fig 7B), supporting the difference observed between the S1068 and A1068 recombinant proteins.

### Generation and phenotype of a transgenic parasite line expressing VAR2CSA S1068A

Both Human and *P. falciparum* CK2α phosphorylate S1068, and its mutation to alanine slightly reduces rDBL1-6 *in vitro* CSA binding while its phosphorylation enhances it. To assess the role of S1068 in IEs cytoadhesion, CRISPR-Cas9 gene editing was performed to introduce the S1068A substitution into the parental NF54 line (see Materials and methods and S7 Fig). We blasted genomic regions flanking PAMs corresponding to the guide sequence against the whole *P. falciparum* genome for potential off-target. We did not find evidence of any homologous sequences besides the gene of interest. Several clones were produced by limiting dilution. The presence of the expected mutation and the absence of additional mutations in the var2csa locus were then confirmed by genomic DNA sequencing of one of the clones named S1068A E9 (S7 Fig).

The *var* gene transcription profile shows 90% of *var2csa* transcripts by qPCR (normalized to housekeeping genes), confirming that this *var* gene is preferentially expressed in this transgenic cell line (Fig 8A). However, flow cytometry experiments using a specific VAR2CSA antibody revealed a highly reduced level of VAR2CSA at the IEs surface, suggesting impaired translocation. In contrast, 80% of the parental wild-type parasite line displays VAR2CSA protein on the surface of infected erythrocyte (Fig 8B). This observation was

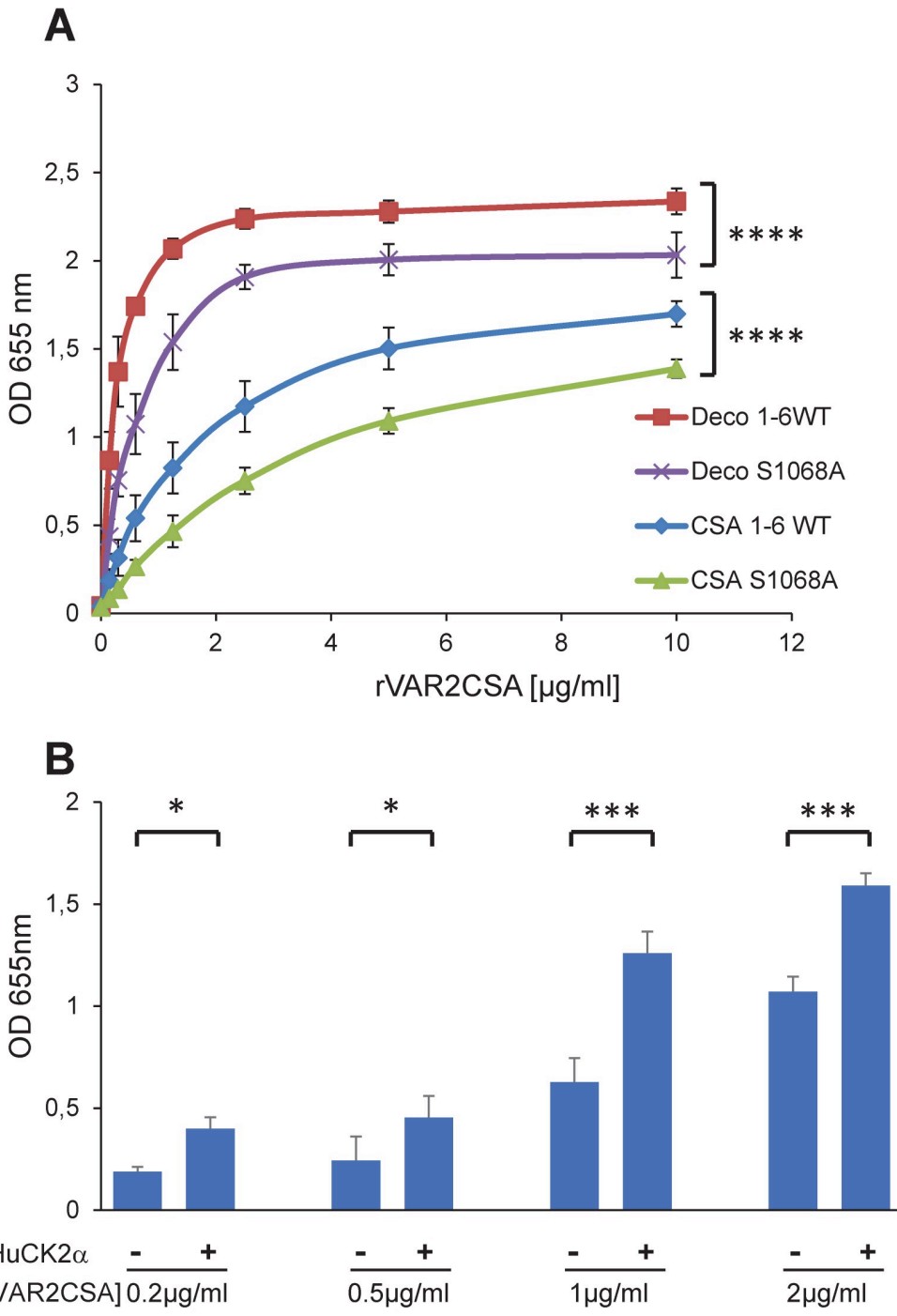

**Fig 7. rDBL1-6 S1068A mutation impairs CSA-binding while its phosphorylation by rHuCK2α enhances CSA binding. (A)** Wild-type DBL1X-6 (1-6WT) and mutated DBL1X-6 S1068A (S1068A) recombinant proteins were assayed by ELISA at different protein concentrations for *in vitro* binding to CSA or decorin coated ELISA plates. Increasing concentrations of recombinant DBL1X-6ε proteins at serial dilutions of 0.156 to 10 μg/mL were added to wells previously coated with CSA, or decorin. Error bars correspond to SD between 3 independent experiments. Each experiment was performed in triplicate. P values were determined by running an unpaired t test on the two area-under-the-curve (AUC) according to the method described in the following web link: https://www.graphpad.com/support/faqid/2031/. **(B)** Human casein kinase 2 plus ATP increases binding of VAR2CSA to CSA. 0.2, 0.5, 1 and 2 μg/mL of VAR2CSA recombinant proteins were preincubated 30 min at 30˚C in kinase buffer supplemented with 10μM ATP with 300 ng or without Human CK2α kinase and added to wells previously coated with CSA. Error bars correspond to

SD between 3 independent experiments. Statistics (paired t test; *p = 0.018; *p = 0.03; ***p = 0.0008 and ***p = 0.0002 for respectively 0.2, 0.5, 1 and 2μg/ml rDBL1-6). CSA, chondroitin sulfate A; Deco, decorin; ELISA, enzyme-linked immunosorbent assay; OD, optical density at 655nm; SD standard deviation.

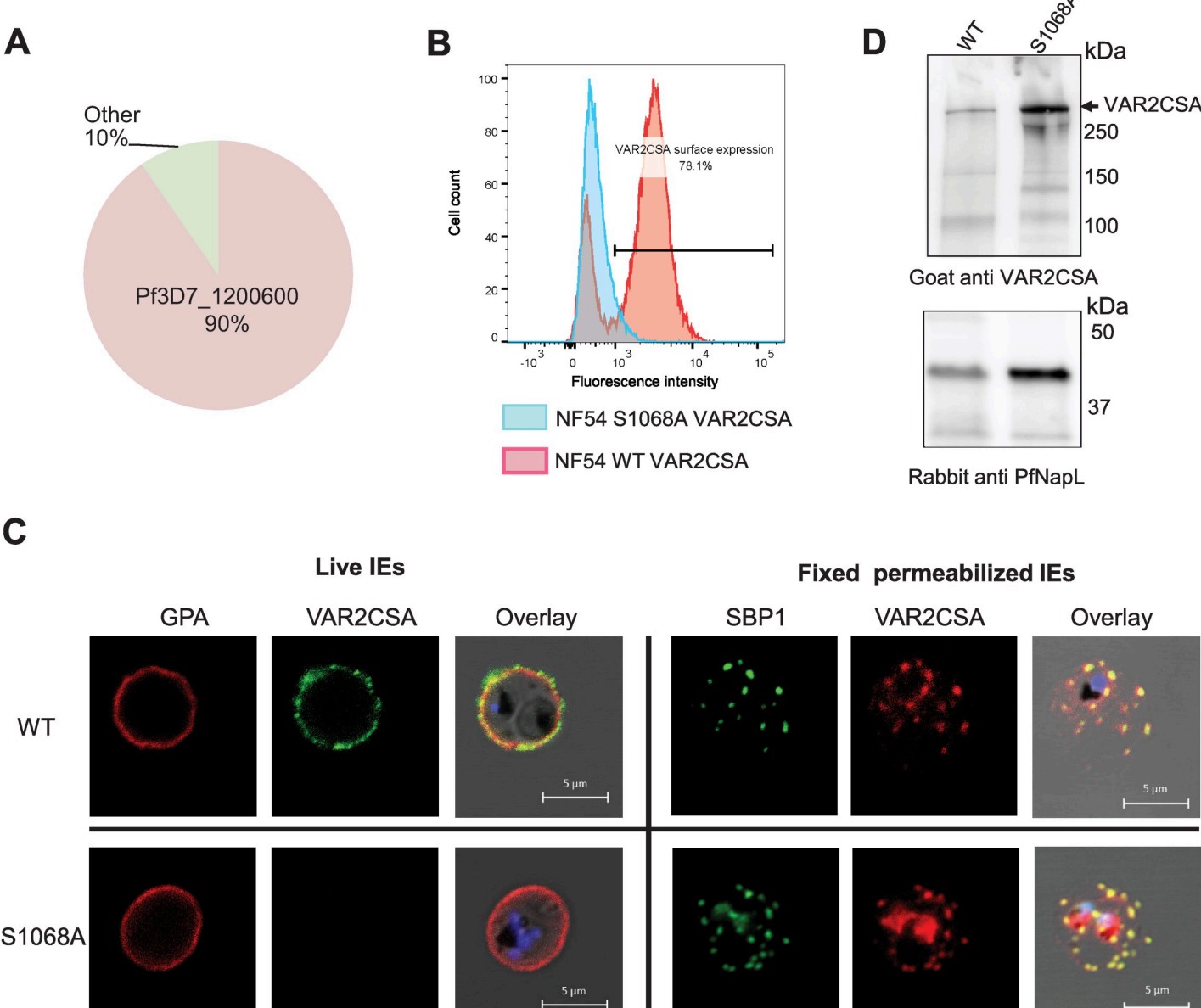

**Fig 8. Phenotype of the transgenic NF54CSA S1068A line. (A)** *Var2csa* transcriptional profile of S1068A shown by qPCR. Transcriptional levels of each *var* genes were normalized with the housekeeping gene, seryl-tRNAtransferase. **(B)** Flow cytometry analysis of wild-type and S1068A NF54CSA IEs. IEs were labelled with rabbit anti-VAR2CSA antibodies. Geometric means of fluorescence intensities and percentage of IEs expressing VAR2CSA are indicated. **(C)** VAR2CSA immunofluorescence assays (IFA). IFA staining was performed on live cells and fixed and permeabilized smears on wild-type NF54CSA IEs and transgenic NF54 S1068A IEs with rabbit anti-VAR2CSA, anti-GPA and anti-SBP1 antibodies. **(D)** Western blot analysis with a goat anti-VAR2CSA antibody was performed on total lysates of wild-type NF54CSA and NF54 S1068A IEs (Upper panel). An anti-PfNapL (PfNucleosome Assembly protein L) serum was used as a loading control for both lysates (Lower panel).

verified by live IFA performed with a rabbit anti-VAR2CSA antibody and a marker of the erythrocyte membrane, Glycophorin A (GPA), (Fig 8C). Surface labelling of the RBC membrane showed that the typical punctuated pattern of VAR2CSA clearly colocalizes with GPA in the wild-type, whereas no VAR2CSA was detected on the surface of the mutant cell line. The same results have been obtained with a second, mutant clone (S10 Fig). IFA performed on fixed and permeabilized IEs with an anti-Skeleton Binding Protein 1 (SBP1) antibody to visualize the Maurer's Clefts, an intermediate trafficking compartment for PfEMP1 translocation to the RBC membrane, showed that in both cell lines VAR2CSA and SBP1 display the same staining pattern, suggesting that VAR2CSA is present in this intermediate organelle but its export to the IEs surface is impaired in the mutant cell line. To further confirm the expression and integrity of endogenous VAR2CSA, total IEs lysates were prepared from the NF54CSA WT and NF54 S1068A cell lines and loaded on an SDS-PAGE followed by western blot analysis (Fig 8D). The size of the protein in S1068A mutant transgenic line is similar to that of wild-type parental line. An anti-PfNucleosome Assembly protein L (PfNapL) serum was used as a loading control for both lysates. No significant difference in the level of VAR2CSA between the WT and S1068 IEs extracts was observed after quantification of the VAR2CSA and PfNapL bands intensities (S8 Fig). Indeed, the signal intensities of the VAR2CSA bands were 10-fold higher in the S1068A IEs extract in comparison to the VAR2CSA WT IEs extract, while a 8.5-fold change was observed for the reference loading control PfNapL. As expected, no adhesion of NF54 S1068A was observed in the static binding assay with immobilized CSA (S9 Fig). We assessed the phenotype of another S1068A clone (A10). Using several approaches (qPCR, cytometry, IFA and IEs cytoadhesion) this mutant reveals the same phenotype as the E9 clone with the lack of VAR2CSA on the IEs surface (S10 Fig). To ensure that the S1068A transgenic parasites PfEMP1 export machinery to the IEs surface was still fully functional, we verified the presence of the *kharp* locus associated to knobs formation in both E9 and A10 clones and performed 2 rounds of S1068A IEs E9 clone panning on CD36-expressing CHO745 cells (S11 Fig). No *kharp* locus deletion was observed in both mutant cell lines. Furthermore, parasites selected for CD36 adhesion, displayed a decreased expression of *var2csa* (90% to 57%) while expression of multiple group B and C *var* genes was observed, indicating that *var* gene switching is not impaired and that CD36-binding PfEMP1 can be exported to the IEs surface. IEs cytoadhesion to CD36 receptor was also verified by static binding (S11 Fig). Taken together these results indicate that the S1068A E9 clone has an intact PfEMP1 export machinery.

## Discussion

VAR2CSA is the sole variant PfEMP1 family member responsible for IEs cytoadhesion to CSA in the placenta and stands as the leading vaccine candidate to protect pregnant women against PM [10]. VAR2CSA is a 350 kDa transmembrane protein with a 300 kDa extracellular region, including the minimal CSA binding domain located in the N terminal region of the protein [33]. Reversible phosphorylation/dephosphorylation of amino acids in the extracellular domains of cell surface proteins is emerging as a critical regulatory mechanism in health and diseases, and ultimately controls many physiological processes such as extracellular signaling, cell polarity, cell adhesion and cell-cell interaction [34]. Phosphorylation of these ectodomains can occur intracellularly during biosynthesis, trafficking to the cell surface or after translocation by membrane-bound kinases or ectokinases [35]. Many phosphorylated proteins have been identified at the plasma membrane [36], but little is known about how the phosphorylation of extracellular domains of proteins might affect cell functions such as cell adhesion. Interestingly, our previous published studies showed that the extra domain of VAR2CSA is

phosphorylated (mainly in the CSA binding domain), and this phosphorylation increases the adhesive properties of IEs to placental CSA. Over the last decades, an increasing number of studies have pointed out that the essential protein kinase CK2 is a "master regulator" of several signaling pathways implicated in the pathogenicity of several diseases [24,37]. This pleiotropic function of CK2 is in line with its localization in a majority of cell compartments [19], including as an ectokinase on the outer surface of the plasma membrane [36]. PfCK2α, the *Plasmodium* ortholog, is essential for the survival of the parasite; it is expressed at all stages, with greater expression in mature stages, and has been shown to be present in the nucleus and cytoplasm [27,28].

Although the phosphorylation of the cytoplasmic domain of the antigenic variant PfEMP1-VAR2CSA by host CK2 has already been shown using CK2α inhibitors [18], we report here that two of these compounds, DMAT and TBCA, drastically reduce the phosphorylation of the extracellular domain. We cannot exclude that this effect may be mediated indirectly (for example, by kinases regulated by CK2α). We demonstrate here that both native and recombinant Human and *Plasmodium* CK2α can phosphorylate the extracellular region of VAR2CSA. Interestingly, although the prediction of phosphorylation motifs for *Plasmodium* CK2 is still unknown, our mass spectrometry analyses identified in the CIDR domain a site (S1068 -D-N-E) which is phosphorylated by both kinases and is compliant with the established Human CK2 recognition motif (S/T-X-X D/E). However, the neighboring serine at position 1059, although located within an acidic environment, was not phosphorylated by Human CK2α. This confirms that S1068 is the dominant CIDR target for the host enzyme. Additional phosphosites, such as S433, which we have previously shown to be important for CSA binding and is a predicted Human CK2 phosphosite [17], have been identified as targets for the *Plasmodium* orthologue, suggesting different phosphorylation consensus between the two kinases and, therefore a differential mode of action.

We focused our interest on the S1068 since both enzymes phosphorylate this residue. Substitution of S1068 to alanine leads to a slight reduction of *in vitro* rVAR2CSA binding to CSA and decorin, suggesting a contribution of this phosphosite in VAR2CSA interaction to CSA. Furthermore, a transgenic parasite line carrying the S1068A mutation displays impaired surface translocation. Whether phosphorylation of this site occurs during the trafficking pathway and/or in at the cell surface is still unknown, but this result is in line with our previous data showing that proper trafficking of VAR2CSA depends on the phosphorylation status of another phosphosite at position T934 [17]. The S1068 phosphorylation status could temporally and spatially controls the trafficking of VAR2CSA. Interestingly, it has been recently shown in another system that phosphorylation by Human CK2 is necessary for sodium channel trafficking to the plasma membrane and channel activity [38]; suppression of the CK2 phosphorylation site by substitution to alanine in the protein impedes channel trafficking to the membrane and decreases channel activity [38]. Extracellular phosphorylation of membrane-bound proteins such as receptors, complement components or transmembrane proteins by CK2 ectokinase activity has been suggested several years ago [39–42]. Indeed, Human CK2α is located in several cell compartments, including the outer leaflet of the plasma membrane [36]. PfCK2 is expressed at all erythrocytic stages, with greater expression in mature stages, where it is present in the nucleus and cytoplasm [27,28]. Furthermore, four PfCK2α peptides were found in a proteomic study aiming to identify IEs surface proteins, using surface biotinylation [43]. Therefore, PfCK2 might be exported. We cannot exclude that PfCK2α is a true exokinase released in the extracellular environment upon physiological stimuli and may phosphorylates VAR2CSA during its export to the IEs surface or in the extracellular medium.

We report that DMAT and TBCA impair IEs cytoadhesion without affecting VAR2CSA translocation, raising several hypotheses. One possibility is that other kinases cooperate to

phosphorylate S1068 in the presence of CK2 inhibitors. Also, it is possible that the pool of the CK2 kinases is not fully inhibited by these molecules, hence, some CK2 enzymes are still active and phosphorylate this residue. Although DMAT and TBCA are known permeable inhibitors [18], these molecules could inhibit in priority the kinase activity of the ecto-CK2 pool and possibly secreted CK2 (exokinase) reducing phosphorylation of VAR2CSA expressed on the surface and subsequently IEs cytoadhesion. Furthermore, before being exposed on the RBC surface, PfEMP1 must be exported beyond the confines of the parasite itself, crossing first the parasite plasma membrane and then the parasitophorous vacuole membrane, and finally transiting through the RBC cytoplasm in the Maurer's clefts structure before being translocated to the RBC surface [44]. It is possible that some of these compartments are not permeable to these inhibitors and that only VAR2CSA present on the IEs surface is affected by the treatments.

Taken together, these results demonstrate that host and *P. falciparum* CK2 phosphorylates the extracellular region of VAR2CSA, and that this post-translational modification is important for proper VAR2CSA trafficking and IEs cytoadhesion to CSA. These findings provide detailed insights into the regulation of IEs sequestration in the placenta and point to a possible immunotherapeutic approach for placental malaria, based on monoclonal antibodies targeting phosphosites that mediate VAR2CSA-dependent cytoadhesion.

## Material and methods

### Cloning and generation of constructs for *P. falciparum* transfection

Vector pUFCas9 was a gift from JJ Rubio Lopez. The design of the pL7-Var2CSA S1068A plasmid was based on the previously described method [17]. We generated a pL7-Var2CSA S1068A plasmid bearing a sgRNA targeting the *var2csa* locus, a homology region or donor sequence and the WR99210 drug-selectable marker hDHFR. This pL7-Var2CSA S1068A plasmid was co-transfected with the pUF1-Cas9 episome carrying the Cas9 endonuclease (S7 Fig). All primers used in the present study are listed in Table 2. Using primers P1 and P2 encoding the *var2csa* gRNA targeting sequence for cas9, we cloned the PCR product in the pL7 vector digested by BtgZ1. A donor fragment of approximately 500bp of the Var2CSA homology

**Table 2. Oligonucleotides used in this study.**

| P1 | seed S1068Fo | taagtatataatattctttcattgtcactaaatttgttttagagctagaa |
|----|--------------|-------------------------------------------------|
| P2 | seed S1068 Rev | ttctagctctaaaacaaatttagtgacaatgaaagaatattatatactta |
| P3 | ext Var2S1068A mutdonor Fo | aatggcccctttccgcggctacgaattatgtaaatataatggtgtag |
| P4 | ext Var2 S1068A mutdonor Rev | ttttacaaaatgcttaaggtatatattcttcccaacatgaaaaaaac |
| P5 | int S1068A mutFo | gcgcaaccaaagtttgctgacaatgaaag |
| P6 | int S1068Amut Rev | ctttcattgtcagcaaactttggttgcgc |
| P7 | Var2CSA S1068AFo | gcctaagttcgccgataatgagag |
| P8 | Var2CSA S1068ARe | ctctcattatcggcgaacttaggc |
| P9 | Var2CSA1059Fo | gtctcgggtgggcgatgaagccgcc |
| P10 | Var2CSA1059Rev | ggcggcttcatcgccacccgagac |
| P11 | CIDR Fo | ggcaggaattcaatgaagaaacctgtgatgac |
| P12 | CIDR Rev | ggcaggcggccgctcaatgatgatgatgatgatggctggtctcagagcttttca |
| P13 | INT1CIDR Fo | tgacgaattcaataaagcctgtattacccacagc |
| P14 | INT1CIDR Rev | gctt gtcgacttaatgatgatgatgatgatggctggtctcagagcttttcatc |
| P15 | KHARP Fo | gcacaccaccatcatcatgga |
| P16 | KHARP Rev | cgtgtgcacttcctccataagca |

containing the Sac2/Afl2 ends plus the 15 bp necessary for infusion cloning in the pL7 vector was amplified by PCR using the external primers P3 and P4 and internal primers P5 and P6 bearing the desired mutations plus the shield mutation. Shield mutation protects the modified locus from repeated cleavages by casp9 enzyme. All PCR products were amplified with high-fidelity polymerase Pfu Ultra II Fusion DNA polymerase (Agilent Technologies).

## Plasmid constructs for recombinant proteins

The CIDR domain and multidomain INT1CIDR of VAR2CSA sequence (Plasmo DB accession number PF3D7_1200600) were PCR amplified from synthetic *var2csa* gene and cloned into the pET24a vector respectively between the EcoR1/Not1 and the EcoR1/Sal1 restriction sites, in frame with the C-terminal hexa-His tag, using the primers P11 and P12 for CIDR or P13 and P14 for INT1CIDR sequences. Similarly, genes encoding 3D7-DBL2X and 3D7-DBL1X-2X were PCR amplified and cloned into a modified pET21b vector in frame with a C-terminal hexa-His tag with specific primers as already described [33]. The gene encoding FCR3-DBL3X (residues 1218–1577), cloned into a modified pET15b vector, was a kind gift from Dr. Matthew K. Higgins. Expression was carried out as previously described [33]. Site-directed mutagenesis was performed using the Quick-change II XL kit from Agilent according to the manufacturer's protocol. Primers with the desired point mutations used in this study are listed in the primers table. Point mutation S1068A was introduced using pTT3-DBL1-6 and pET24a-CIDR wild-type vectors as matrices and primers P7 and P8. Point mutation S1059G was introduced with primers P9 and P10 in pET24a CIDR vector. The presence of mutations was verified by sequencing before protein expression. The double mutant pTT3-S429A/S433A construct was previously described [17].

## Expression and purification of recombinant fusion proteins

DBL1X-3X, DBL4ε-6ε and DBL1X-6ε Var2CSA were cloned into pTT3 vector and expressed in HEK293-F (embryonic human kidney) [33] as soluble proteins secreted in the culture medium. Proteins were purified on a His-Trap Ni affinity column, followed by an ion exchange chromatography (SP Sepharose) and a gel filtration chromatography (Superdex 200) according to the protocol previously described [33]. The amino acids from N962 to S1209 (CIDR) and amino acids from N445 to S1209 (INT1 CIDR) of VAR2CSA protein sequence from the synthetic gene [33] were expressed in the *E. coli* bacterial Shuffle strain as cytoplasmic soluble proteins and purified accordingly to the Material and Methods as described previously [33]. GSTPfCK2α or HisPfCK2α expression was performed in Rosetta cells in LB media supplemented with 100 μg/ml Ampicillin and 34 μg/ml Chloramphenicol overnight at 20°C after induction with 0.1mM IPTG. The purification protocol was performed as reported previously [45]. Human CK2α, fused or not to MBP, was purchased at Biaffin (Ref: PK-CK2AH-A010; PK-CK2AH2-A010). MBP alone was expressed from the pMAL vector (NEB #N8108) and purified according to the manufacturer's recommendations.

## Parasite culture and parasite transfection

*P. falciparum* FCR3CSA, NF54CSA, and FCR3CD36 strains were maintained in culture under standard conditions in O$^+$ Human erythrocytes in RPMI 1640 containing L-glutamine (Invitrogen) supplemented with 5% Albumax I, 1 × hypoxanthine and 20 μg/mL gentamicin. CD36 or CSA-binding IEs phenotypes were verified on receptors immobilized on plastic Petri dishes as previously described [15]. IE cultures (3–5% parasitemia) at mid/late trophozoite stages were purified using the VarioMACS system with CS columns (Miltenyi) [33]. Genomic DNA extracted from parasite cultures was regularly tested for Mycoplasma contamination

(look out Mycoplasm PCR detection kit (Sigma)) using MSP primers [46]. All transgenic *P. falciparum* parasites were generated from *P. falciparum* NF54CSA strain. Parasites were transfected at the ring stages by electroporation. 50 μg of each plasmid pL7-Var2CSA S1068A and pUF1 Cas9 were used for each transfection after ethanol precipitation and resuspension in sterile TE. Both drugs: WR99210 (2.6nM) and DMS1, a DHODH inhibitor, 5-Methyl-N-(2-naphthyl) (1,2,4) triazolo(1,5-a) pyrimidin-7-amine (1.5μM) were added 20 hours post-transfection and applied every day. The apparition of resistant parasites was monitored by Giemsa staining.

## *Var* gene expression analysis by quantitative PCR (qPCR)

RNA from transgenic NF54CSA synchronized ring stages parasites was extracted with TRIzol following the manufacturer's instructions (Qiagen). cDNA synthesis was performed by random primers after DNase I treatment (TURBO DNase, Ambion) using the SuperScript III First Stand Synthesis system (*Invitrogen*). Primer pairs used to detect each *var* gene expression have been described previously [47]. Real-time PCR reactions were performed on a CFX 96 thermocycler (Biorad). Transcriptional level of each *var* gene was normalized with the housekeeping control gene seryl tRNA transferase (PlasmoDB: PF3D7_0717700).

## CD36 selection by successive pannings

IEs were enriched for CD36 binding on CD36-expressing CHO745 cells. VarioMACS purified IEs at trophozoites stages were allowed to attach to CHO cells in binding buffer (RPMI 1640 with HEPES and without NaHCO$_2$, 10% albumax at pH 6.8) for 30 min, mixing every 10 min. Unbound IEs were washed out 5 times with RPMI media by rocking the flasks. Bound parasites were removed from the CHO cells by flushing media with a pipette. Parasites were put back in culture with fresh blood. Smears were performed routinely to check for apparition of IEs. *Var* gene expression profiling was performed as described above after two rounds of panning.

## Flow cytometry analysis

Wild-type and transgenic NF54CSA infected erythrocytes at mid/late trophozoites stages were resuspended in PBS 1% BSA after VarioMACS purification and counted. For each assay, 3 x10$^5$ IEs were washed in PBS and incubated with 50μl of purified rabbit anti-VAR2CSA antibody diluted 1:100 in PBS 1% BSA for 1 h at RT. IEs were washed twice with PBS and resuspended in 100μl of PE-conjugated goat anti-rabbit antibody diluted 1:100 in PBS for 30 min at RT. After washing twice in PBS 1% BSA, IEs were resuspended in paraformaldehyde 4% in PBS and kept at 4˚C overnight in darkness. Cells were washed twice with PBS and analyzed by flow cytometry using a BD FACS canto II or a LSR Fortessa cytometers. The results were analyzed using the FlowJo 10.0 software. Parasite nuclei were stained by TO-PRO-3 (1:10,000 dilution). The results shown are geometric mean fluorescence intensities.

## ELISA binding assay

ELISA plates were coated with 1mg/mL of chondroitin sulphate A (CSA) (Sigma, C8529) in PBS (Gibco, NaCl 150 mM pH 7.2), or decorin (5μg/ml) using 100 μL per well overnight at 4˚C.

Wells were blocked with 150 μL of PBS 1% BSA buffer per well 2 h at 37˚C. After removal of the blocking solution, 100 μl of serial dilutions (from 10 μg/ml to 0.156 μg/ml) of the recombinant VAR2CSA protein (wild type and mutated) in PBS 1% BSA, 0.05% Tween 20 were added per well and incubated for 2 h at 37˚C. For binding assays carried with the rHuCK2α

(300ng), dilutions of rDBL1-6 protein were preincubated in cold kinase assay buffer plus ATP 10µM for 30 min at 30°C, diluted in PBS 1% BSA prior addition into the wells. After washing twice with PBST (PBS plus 0.05% Tween 20), 100 µL anti-His HRP-conjugated antibody (diluted 1:3000 in PBST) or a polyclonal rabbit anti-VAR2CSA (1:1000e) followed by a goat anti-rabbit conjugated to HRP (diluted 1:3000) was added to each well and incubated for 1 h at 37°C. Again, after washing thrice with PBST, the reaction was revealed with 100 µL per well of substrate (TMB: 3,3',5,5'-tetramethylbenzidin; Biorad) until saturation was reached. Interaction was related to the absorbance monitored at 655 nm.

## Static adhesion assays on immobilized receptors

20µl of CSA (1mg/ml) in PBS or CD36 (50µg/ml) in PBS or BSA 1% in PBS (negative control) were spotted on a Petri dish (approximately 0.5 cm diameter circles) overnight at 4°C in a humidified chamber and used for static binding assays with infected erythrocytes. The spots were washed twice in PBS and blocked with PBS BSA 1% for 1 h at RT. Mature stages were purified using VarioMACS and $10^5$ IEs were added to the spotted plates at RT for 1 h. For experiments with CK2α inhibitors, ring stages or mid/late stages were treated for 16 hours or 1 hour, respectively with 50µM of TBCA or DMAT. DMSO was used as a negative control in each experiment. Mature stages were purified using VarioMACS and $10^5$ IEs were added to the spotted plates and the static assays were carried as described above. Unbound IEs were gently washed away twice with PBS. Adherent infected red blood cells were fixed with 2% glutaraldehyde and counted on 5 fields in duplicate spots with a Nikon Eclipse Ti microscope with a 10 X objective. Results are expressed as the binding percentage compared to 100% binding of the positive control.

## Preparation of total uninfected or infected erythrocytes extract

Total protein extracts were prepared from uninfected red blood cells (uRBC) or from asexual mixed stages NF54CSA parasites or transgenic NF54 harvested after MACS purification. After washing with cold PBS, the parasites pellets were resuspended in cold lysis buffer (150mM NaCl, 50mM Tris HCl pH 8.0, 1% NP40, 0.5% Na Deoxycholate, 0.05% SDS supplemented with proteases and phosphatases inhibitors) incubated on ice and sonicated briefly. The lysates were cleared by centrifugation (20,000 x g for 20 min at 4°C), the total amount of proteins in the supernatant was measured using the Bradford assay and collected for immunoprecipitation or western blot applications.

## Preparation of membrane fraction lysates of uninfected and infected RBC

uRBC or IEs soluble and membrane fractions were prepared as described previously [48]. Briefly, the infected and uninfected cells were resuspended in NETT lysis buffer (150mM NaCl, 5mM EDTA, 50mM Tris HCl pH 8.0, 1% Triton X-100) supplemented with protease and phosphatase inhibitors. The membrane fraction was separated from soluble cytosolic material by centrifugation at 20,000 x g 4°C for 30 min. The pellet was resuspended and dissolved at room temperature in Tris saline buffer (pH8) supplemented with SDS 2%, proteases and phosphatases inhibitors. After centrifugation at RT at 20,000 x g, the resulting supernatant corresponds to solubilized membrane fraction.

## Immunoprecipitation

Membrane fractions prepared above were diluted ten to fifteen times in NETT buffer prior to immunoprecipitation to wash away the SDS. For immunoprecipitation experiments,

erythrocytes membrane fractions lysates were incubated for 2 h with 3μg of mouse IgG isotype or mouse immuno-purified anti-HuCK2α (*Santa Cruz Biotechnologies*). Total lysates (500μg) of infected RBC were incubated for 2 h at 4˚C with 3μg of immunopurified rabbit anti-PfCK2α or pre-immune immunopurified rabbit antibody. Immunoconjugated material was precipitated with 20μl of Protein G Sepharose after centrifugation, washed four times in NETT buffer and recovered by heating samples 3 min at 95˚C for western blots applications. For kinase assays, the immunoprecipitated proteins bound to the beads were washed once with kinase assay buffer and used as a source of kinase in standard phosphorylation assays. Poly-clonal anti-VAR2CSA (3μg) was used to immunoprecipitate endogenous VAR2CSA from the membrane fraction of red blood cells infected by various *P. falciparum* strains, as reported previously [48]. Immunoprecipitated VAR2CSA attached to the beads was used as a substrate in standard phosphorylation assays.

## Western blotting

Anti-HuCK2α western blot was performed using a goat immuno-purified antibody (Santa Cruz; 1:200 dilution following manufacturer's recommendations) followed by a secondary rabbit anti-goat antibody conjugated to peroxidase (1:3000). For PfCK2α, western blot was carried out using a rabbit anti-PfCK2α at 1:1000 followed by incubation with a secondary goat anti-rabbit antibody conjugated to peroxidase (1:3000). A mouse monoclonal anti-VAR2CSA (1μg/ml) or a goat polyclonal anti-VAR2CSA (1:1000) were used, followed respectively by a goat secondary anti-mouse antibody or by a secondary rabbit anti-goat antibody both conjugated to peroxidase (1:3000). Recombinant VAR2CSA DBL 1–6 was used as a control. His-tagged proteins were detected with a mouse anti-His HRP-conjugated antibody (1:3000) from Qiagen. MBP fusion proteins were detected with a rabbit anti-MBP from NEB company 1:1000 followed by an incubation with a goat anti-rabbit HRP-conjugated (1:3000). A mouse anti-GST antibody (Qiagen) diluted (1:1000) was used to detect GST tagged proteins followed by a secondary goat anti-mouse antibody conjugated to peroxidase at (1:3000). anti-PfNapL serum was a gift from C. Doerig's lab and was used at a dilution of 1:1000.

## Immunofluorescence assays

IFA on live and fixed mature-stage parasites was carried out as followed. Live mature-stage parasites were incubated 1 hour on ice with a rabbit anti-VAR2CSA polyclonal immunopurified antibody (dilution 1:500) and a monoclonal mouse anti-Glycophorin A (BDMAB1228) (dilution 1:400). After 2 washes in PBS 1% BSA, the parasites were labelled 45 min on ice with a donkey anti-rabbit Alexa Fluor 488 (Invitrogen A-21206) diluted 1:400 and a goat anti-mouse Alexa Fluor 568 (Invitrogen A11031) diluted 1:400 on ice. After Hoechst staining, the parasites were observed with a Zeiss LSM700 confocal microscope under a 63X immersion oil objective. For IFA on fixed and permeabilized cells, smeared parasites were fixed in cold acetone/methanol (90/10) for 15 min. After blocking with 1% BSA in PBS for 1h, the slides were incubated with a rat anti-SBP1 serum (dilution 1:300) and a goat anti VAR2CSA (dilution 1:500) followed by incubation with a donkey anti-rat Alexa Fluor 488 diluted 1:400 and a donkey anti-goat Alexa Fluor 568 (Invitrogen A11057) diluted 1:400. After washes, the slides were mounted with Fluoromount plus DAPI (4',6-diamidino-2-phenylindole) and observed as described above.

## Kinase assays

Immunoprecipitated VAR2CSA, recombinant full length, single or multi-domain VAR2CSA were used as substrates. Protein lysates, recombinant or immunoprecipitated kinases were used

as a source of kinase in a standard kinase reaction assay with 10μM ATP, 2.5 μCi $^{32}$P γATP in kinase buffer (20mM Tris pH 7.5, 20 mM MgCl$_2$, 2mM MnCl$_2$) supplemented with phosphatases inhibitors. Reactions were carried out at 30˚C for 30 min and stopped with loading dye sample, boiled at 95˚C and loaded onto SDS PAGE. Gels were stained, destained and exposed for autoradiography. Non-radioactive kinase assays were carried out using cold ATP only.

### Interaction assay

A mixture of 5μg of each recombinant protein was incubated at 4˚C for 30 min in 20mM Tris-HCL (pH 7.5), 0.2MNaCl, 0.1% NonidetP40 (IGEPAL) and 10% glycerol. Glutathione-agarose beads or amylose resins were added to each reaction mixture. The tubes were rotated at 4˚C for 1 hour; the beads were recovered by centrifugation and washed four times in reaction buffer. Laemmli buffer was added to the beads and heated at 95˚C. Samples were separated by SDS-PAGE on 4-15% acrylamide stained-free gels prior western blotting.

### Mass spectrometry

**Sample preparation.** Each sample was diluted in 50 μl of 4 M urea + 10% acetonitrile and buffered with Tris-HCl pH 8.5 at a final concentration of 30 mM. Reduction was performed with 10mM dithioerythritol at 37˚C for one hour with constant shaking (600 rpm). Samples were again buffered to pH 8.5 with Tris pH 10–11 and alkylation was performed with 40 mM iodoacetamide at 37˚C for 45 min with constant shaking in a light-protected environment. Reactions were quenched by the addition of dithioerythritol to a final concentration of 10 mM. Samples were then diluted five-fold with 50 mM ammonium bicarbonate and protein digestion was performed overnight at 37˚C using mass spectrometry grade Trypsin Gold (1:50 enzyme-protein) and 10 mM CaCl2 or using chymotrypsin. Reactions were stopped by the addition of 2 μl of pure formic acid. Peptides were desalted on C18 StageTips [49]. Eluted peptides were either dried by vacuum centrifugation prior to LC-MS/MS injections or submitted to phosphopeptide enrichment. Selective enrichment of phosphopeptides were performed on homemade titania tips [50]. Prior to sample loading, the titania tips were equilibrated with 0.75% TFA, 60% acetonitrile, lactic acid 300mg/ml (Solution A). The digested peptides were resuspended in 20 μl of solution A and loaded on a titania tip. After a successive washing step with solution A and 0.1% TFA, 80% ACN (solution B), two elutions were performed with 0.5% ammonium hydroxide and 0.5% piperidine. Eluted fractions were acidified with FA (Formic Acid) and dried in a speedvac.

**LC-MS/MS.** Samples were resuspended in 2% acetonitrile, 0.1% FA and nano-flow separations were performed on a Dionex Ultimate 3000 RSLC nano UPLC system (Thermo Fischer Scientific) on-line connected with an Orbitrap Elite Mass Spectrometer (Thermo Fischer Scientific). A homemade capillary pre-column (Magic AQ C18; 3 μm to 200 Å; 2 cm × 100 μm ID) was used for sample trapping and cleaning. A C18 tip-capillary column (Nikkyo Technos Co; Magic AQ C18; 3 μm to 100 Å; 15 cm × 75 μm ID) was then used for analytical separations at 250 nl/min over 75 min using biphasic gradients. Samples were analysed in data-dependent acquisition mode with a dynamic exclusion of 40 sec. The twenty most intense parent ions from each MS survey scan (m/z 300–1800) were selected and fragmented by CID (Collision Induced Dissociation) into the Linear Ion Trap. Orbitrap MS survey scans resolution was set at 60,000 (at 400 m/z) and fragments were acquired at low resolution in centroid mode.

### Data processing

Raw data were processed using SEQUEST, MS Amanda and Mascot in Proteome Discoverer v.2.4 against a concatenated database consisting of the UniProt human reference proteome

(Release 2014_06) and VAR2CSA sequence. Enzyme specificity was set to trypsin or chymo-trypsin and a minimum of six amino acids was required for peptide identification. Up to two missed cleavages were allowed. For the search, carbamidomethylation was set as a fixed modification, whereas oxidation (M), acetylation (protein N-term) and Phosphorylation (S, T, Y) were considered as variable modifications. Data were further processed using X! Tandem and inspected in Scaffold 5.1 (Proteome Software, Portland, USA). Spectra of interest were manually validated Peptide-spectrum matches with Mascot score > 18 and/or SEQUEST score > 3 were considered as correctly assigned.

## Supporting information

**S1 Fig. VAR2CSA surface expression.** (a) VAR2CSA Surface expression of trophozoite IEs after one treatment with CK2 inhibitors or DMSO, monitored by flow cytometry with a specific anti-VAR2CSA antibody. (b) VAR2CSA surface expression of ring stage IEs after 16-hours treatment with CK2α inhibitors or DMSO monitored by flow cytometry with a specific anti-VAR2CSA antibody. Geometric means of fluorescence intensities of three independent experiments are represented with standard deviations.
(TIF)

**S2 Fig. Quantification of phosphorylation signal.** *In vitro* radioactive [γ—32P] ATP phosphorylation assays of recombinant His tagged VAR2CSA DBL1-6 protein (1μg in all assays) were performed in the presence of total IEs lysates and increasing concentrations of DMAT or TBCA. The phosphorylation signal for each condition was quantified by Image Lab Software and adjusted to reflect a percentage compared to signal obtained in DMSO control condition (100%).
(TIF)

**S3 Fig. PfCK2α antibodies do not cross react with the Human CK2α.** A rabbit polyclonal anti-PfCK2α and a mouse anti-HuCK2α were used in immunoprecipitation experiments performed on total uninfected RBC extracts. A fraction of the immunoprecipitated material was loaded on a stain free gel prior western blot. Upper panels are the stain free gels and lower panels are the western blots. Recombinant MBP-HuCK2α and GST-PfCK2α were loaded as controls. (a) Anti-PfCK2α western blot. Lane 1: uRBC membrane fractions; lane 2: total uRBC lysates. lane 3: MBP HuCK2α; lane 4: GST PfCK2α; lane 6: Immunoprecipitation with a rabbit anti PfCK2α; lane 7: immunoprecipitation with a mouse anti-Human CK2α. (b) Anti-HuCK2α western blot. Lane 1: uRBC membrane fractions; lane 2: total uRBC lysates. lane 3: MBP HuCK2α; lane 4: GST PfCK2α; lane 5: MW; lane 6: Immunoprecipitation with a rabbit anti PfCK2α; lane 7: immunoprecipitation with a mouse anti-Human CK2α; lane 8: MW. (c) Comparison of the N terminal of Pf and Hu CK2α protein sequences. The designed PfCK2α peptide used for rabbit immunization is boxed in red.
(TIF)

**S4 Fig. Radiolabelled phosphorylation of immunoprecipitated VAR2CSA by rHuCK2α.** Immunoprecipitation of VAR2CSA from membrane fractions of uRBC and various selected VAR2CSA strains was followed by a radiolabelled phosphorylation assay with rHuCK2α (lane 1 to 4). The arrow indicates the immunoprecipitated VAR2CSA from the membrane IEs lysates. Lane 1: immunoprecipitation from uRBC membrane lysates; lane 2: immunoprecipitation from membrane lysates of 7G8CSA IEs; lane3: immunoprecipitation from membrane lysates of NF54CSA IEs; lane 4: immunoprecipitation from membrane lysates of FCR3CSA IEs; lane 5: immunoprecipitation from membrane lysates of NF54CSA IEs without rHuCK2α.
(TIF)

**S5 Fig. Expression and kinase activity of GST-PfCK2α and GST-PfCK2αK72M.**
GST-PfCK2α and GST-PFCK2α K72M were expressed in *E. Coli* Rosetta and purified as described in [27]. GST-PfCK2α kinase activity towards rDBL1-6. Autoradiograms (right) and Coomassie blue-stained gels (left) of kinase assays performed with GST-PfCK2α or catalytically inactive GST-K72MPfCK2α. The recombinant kinase and rDBL1-6 substrate are indicated with an arrow. Autophosphorylation of the wild-type kinase is shown.
(TIF)

**S6 Fig. DMAT and TBCA dose dependent inhibition of *in vitro* rDBL1-6 phosphorylation by recombinant kinases.** Recombinant DBL1-6 was used in *in vitro* phosphorylation assays in the presence of [γ—32P] ATP with recombinant *Plasmodium* CK2α or Human CK2α and with increasing concentrations of DMAT and TBCA. (a) (PfCK2α +TBCA): lane1: rDBL1-6 + DMSO; lane2: rDBL1-6 + TBCA 2μM; lane 3: rDBL1-6 + TBCA 10μM; lane 4: rDBL1-6 + TBCA 50μM; (b) (HuCK2α + DMAT): lane1: rDBL1-6 + DMSO; lane2: rDBL1-6 + DMAT 2μM; lane 3: rDBL1-6 + DMAT 10μM; lane 4: rDBL1-6 + DMAT 50μM.
(TIF)

**S7 Fig. CRISPR/Cas9 strategy used for S1068 substitution.** Nucleotide editing using sgRNA: Cas9 in *P. falciparum*. (a) Diagram illustrating the strategy used for nucleotide replacement. The Cas9 protein is expressed in the pUF1-Cas9 episome continuously maintained using the ydhodh drug-selectable marker. PL7- var2csa episome is maintained using the hDHFR selection and carries both the sgRNA var2csa and the donor DNA (blue box). The donor DNA carries the designed desired mutation (red star) and the shield mutation (blue star). (b) sgRNA var2csa targeted sequences recognized by Cas9. The 20 nucleotides guide and PAM sequences are indicated. (c) Chromatograms showing sequence analyses of var2csa locus in NF54CSA wild-type and in transgenic NF54 S1068A E9 clone. Nucleotide substitutions and amino acids changes in var2csa locus are highlighted.
(TIF)

**S8 Fig. Quantification of the VAR2CSA and PfNapL bands.** Quantification of the intensity of the bands of anti-VAR2CSA and anti-PfNapL western blots was performed by ImageJ software.
(TIF)

**S9 Fig. Static adhesion assay.** Image of a field showing bound NF54CSA parental and transgenic NF54 S1068A E9 clone IEs on CSA coated on plastic.
(TIF)

**S10 Fig. Characterization of the clone A10 NF54 S1068A.** (a) Var transcriptional profile of S1068A clone A10 shown by qPCR. Transcriptional levels of each *var* genes were normalized with the housekeeping gene, seryl-tRNA transferase. (b). Flow cytometry analysis of wild-type and S1068A clone A10 NF54 IEs. IEs were labelled with rabbit anti-VAR2CSA antibodies. Geometric means of fluorescence intensities and percentage of IEs expressing VAR2CSA are indicated. (c) VAR2CSA immunofluorescence assays (IFA). IFA staining was performed on live cells on S1068A clone A10 NF54 IEs with rabbit anti-VAR2CSA and anti-GPA. (d) Static CSA cytoadhesion assay of wild-type and S1068A clone A10 NF54 IEs.
(TIF)

**S11 Fig. *Kharp* locus, *var* gene switching and CD36 binding after CHO-CD36 panning of S1068A E9 clone.** (a) *Kharp* locus PCR amplification. PCR was performed using GXL polymerase and *kharp* specific primers on genomic DNA (60ng) of NF54CSA WT parental line and NF54 S1068A mutant clones E9 and A10. Lane1: Clone VAR2CSA S1068A A10; lane2:

Clone VAR2CSA S1068A E9; lane 3: WT parental line; lane 4: $H_2O$ negative control; Lane 5: Molecular Weight. (b) Var transcriptional profile of S1068A clone E9 before and after 2 rounds of panning on CHO cells expressing CD36 shown by qPCR. Transcriptional levels of each *var* genes were normalized with the housekeeping gene, seryl-tRNAtransferase. (c) CD36 cytoadhesion assay of S1068A clone E9 IEs after 2 rounds of panning on CHO cells expressing CD36.
(TIF)

**S1 Data. Source data for graphs in this study.**
(XLSX)

## Acknowledgments

We would like to thank Célia Dechavanne for performing the statistical analysis and Artur Scherf for providing the pL7 plasmid.

## Author Contributions

**Conceptualization:** Dominique Dorin-Semblat, Christian Doerig, Benoit Gamain.

**Data curation:** Dominique Dorin-Semblat, Jean-Philippe Semblat, Romain Hamelin, Anand Srivastava, Marilou Tetard, Benoit Gamain.

**Formal analysis:** Dominique Dorin-Semblat, Jean-Philippe Semblat, Romain Hamelin, Anand Srivastava, Marilou Tetard, Christian Doerig, Benoit Gamain.

**Funding acquisition:** Benoit Gamain.

**Investigation:** Dominique Dorin-Semblat, Jean-Philippe Semblat, Anand Srivastava, Marilou Tetard, Christian Doerig, Benoit Gamain.

**Methodology:** Dominique Dorin-Semblat, Jean-Philippe Semblat, Romain Hamelin, Anand Srivastava, Marilou Tetard, Graziella Matesic, Benoit Gamain.

**Project administration:** Benoit Gamain.

**Supervision:** Dominique Dorin-Semblat, Benoit Gamain.

**Validation:** Dominique Dorin-Semblat.

**Writing – original draft:** Dominique Dorin-Semblat, Jean-Philippe Semblat, Benoit Gamain.

**Writing – review & editing:** Dominique Dorin-Semblat, Jean-Philippe Semblat, Benoit Gamain.

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
