## [Decision Letter · Decision Letter 0]

5 Sep 2024

Dear Dr Gamain,

Thank you very much for submitting your manuscript "Casein Kinases 2-dependent phosphorylation of the placental ligand VAR2CSA regulates Plasmodium falciparum-infected erythrocytes cytoadhesion" for consideration at PLOS Pathogens. As with all papers reviewed by the journal, your manuscript was reviewed by members of the editorial board and by two independent reviewers. In light of the reviews (below this email), we would like to invite the resubmission of a significantly-revised version that takes into account the reviewers' comments.

We cannot make any decision about publication until we have seen the revised manuscript and your response to the reviewers' comments. Your revised manuscript is also likely to be sent to the two reviewers for further evaluation.

Sincerely,

David A. Fidock, Ph.D.

Academic Editor

PLOS Pathogens

Tracey Lamb

Section Editor

PLOS Pathogens

Michael Malim

Editor-in-Chief

PLOS Pathogens

orcid.org/0000-0002-7699-2064

Reviewer's Responses to Questions

**Part I - Summary**

Reviewer #1: The submission from Dorin-Semblat and colleagues describes a series of experiments designed to identify the kinase or kinases responsible for phosphorylating an important virulence factor in the malaria parasite Plasmodium falciparum, a protein called VAR2CSA. The authors also provide evidence for a role of phosphorylation in protein trafficking and transport to the infected RBC surface. VAR2CSA is recognized at the primary protein responsible for pregnancy associated malaria and is a high-profile candidate as a potential vaccine that could protect women from this important complication of infection by P. falciparum. Understanding the role of phosphorylation of the exposed portion of the protein therefore has important implications for both our basic understanding of protein processing and trafficking as well as for possible vaccine development.

The authors have built on previous studies from their group and others that detected phosphorylation of VAR2CSA. Here they provide additional evidence that both human and parasite CK2 can phosphorylate VAR2CSA and they identify the specific residues that serve as substrate for the kinases. They also generate a mutant parasite line in which a specific residue has been altered and show that phosphorylation of this residue is important for translocation of VAR2CSA to the infected cell surface. Overall, the manuscript is well written, and the experiments are well designed. There are however a few instances in which additional controls would make the conclusions more convincing.

Reviewer #2: In this manuscript, Dorin-Semblat et al. report that Casein Kinase 2a (CK-2a) is involved in the phosphorylation of Variant Surface Antigen Var2CSA. Var2CSA is exported by the parasite to the surface of host erythrocyte surface and is involved in P. falciparum sequestration and cytoadherence with chonodritin sulfate A (CSA). The authors demonstrate that both human and parasite Casein Kinase 2a phosphorylate recombinant or immunoprecipitated Var2CSA. The phosphorylation sites for CK2a on Var2CSA were also identified. They also demonstrate that CK2a inhibitors block adhesion to CSA and erythrocyte. These inhibitors also impair the phosphorylation of Var2CSA by the parasite and erythrocyte lysate. Finally, they generate a S>A mutant of one of the phosphorylation sites of VAR2CSA in the parasites and demonstrate that the mutant (S1068A) parasites exhibit reduced binding to CSA.

In my opinion the work, largely lacks novelty as previous studies have already demonstrate that CK2 inhibitors prevent VAR2CSA phosphorylation (Hora et al., JBC 2009). Further there is no direct genetic evidence that PfCK2a indeed phosphorylates Var2CSA in the parasite, which is needed to demonstrate the link between the two as kinase inhibitors can be non-specific. However, experiments performed with S1068A mutant parasites are interesting which seem to suggest that phosphorylation of this site may be important for erythrocyte adherence. Unfortunately, this part of the manuscript is not well-developed.

Specific Comments:

1. Do CK2 inhibitors block parasite growth ?

2. Figures 2-4: Results in these figures demonstrate that human/Pf/CK2 interact with Var2CSA and phosphorylate it in vitro. Several of the experiments demonstrate the same result therefore are repetitive.

The experiments performed with total cell lysates as kinase source are "crude" as several other kinases can phosphorylate VAR2A and the inhibitors can block them especially at high concentrations. Therefore, these may be excluded especially when kinase assays have been performed with purified recombinant hu/PfCk2 or with IP, which provide direct evidence of phosphorylation by the kinase although it is only in vitro.

3. The concentration of TBCA/DMAT used in experiments is in 10-50uM, which seems to be too high. What is the IC50 of these inhibitors against human and PfCK2.

4. Do Var2CSA and CK2- co-immunoprecipitate and co-localize in the parasite and/or IE?

5. Given that the ectodomain of Var2CA is phosphorylated by CK2, how does it come in contact with PfCK2? Does it get phosphorylated in the parasite before it is trafficked to the host ?

6. Figures 5-7: S1068 present in the CIDR is identified as the target phosphorylation site for hu/PfCK2a. There seems to be only a modest decrease in binding of S1068A mutant line to CSA. Statistical analysis of the data provided in Figure 6 should be performed.

7. Figure 7: The finding that trafficking of the S1068A mutant to the host RBC is impaired is interesting. However, more experiments need to establish that S1068A phosphorylation is critical for the export of Var2CSA to the host, which in turn is important for adhesion.

Here are a few suggestions:

a. The quality of IFA provided in Fig. 7c should be better and an additional marker for PVM and RBC surface may be included.

b. Figure 7D Western blot shows that S1068A is expressed at higher levels that WT Var2CSA. Was IE or only parasite lysate was used for this experiment? It may be good to fractionate RBC and parasite fractions and compare the WT and S108A mutant in these fractions.

c. BrefeldinA, which blocks protein transport, can be used to compare the trafficking of mutant and WT Var2CSA. In addition, co-localization with CK2 can be performed.

**Part II – Major Issues: Key Experiments Required for Acceptance**

Reviewer #1: 1. In the section entitled “Endogenous PfCK2α phosphorylates rVAR2CSA”, the authors use antibodies against PfCK2α to immunoprecipitate the protein from infected RBCs, then show that it can phosphorylate recombinant VAR2CSA. There is a formal possibility that these antibodies could also recognize human CK2 or that the human enzyme could otherwise contaminate the precipitated protein, and thus the phosphorylation of recombinant VAR2CSA in the assay could be due to the human protein rather than the parasite protein. While this is unlikely, it could be ruled out by including an additional control of immunoprecipitation from uninfected RBCs.

2. In the section entitled “Effect of S1068A mutation on rVAR2CSA DBL1-6 in vitro binding” the authors perform binding assays with wildtype and mutant VAR2CSA. The authors’ model is that this amino acid is important for phosphorylation, which can affect trafficking, binding or both. Here they investigate a role for binding, but it appears they did not include a kinase to phosphorylate this site. The authors show a slight decrease in binding when comparing unphosphorylated S1068 to A1068, but isn’t the relevant comparison between phosphorylated S1069 to A1068?

3. In the last section of the Results, the authors generate a mutant parasite line in which they have inserted the S1068A mutation and demonstrate that VAR2CSA is not trafficked to the RBC surface. This is a particularly interesting result and adds significantly to the paper. However, it is common for cultured parasites to lose surface expression of PfEMP1, something that is frequently observed in lines of 3D7, for example. It is also worth noting that inhibition of CK2 activity with TBCA or DMAT did not affect surface expression of VAR2CSA. Therefore, validating this phenotype is important. The standard method for validation of a mutant line is via complementation: reverting the mutation and rescuing the wildtype phenotype, in this case VAR2CSA surface expression. An alternative would be to isolate additional clones from an independent transfection and showing the identical phenotype.

Reviewer #2: (From my comments in Part I)

Specific Comments:

1. Do CK2 inhibitors block parasite growth ?

2. Figures 2-4: Results in these figures demonstrate that human/Pf/CK2 interact with Var2CSA and phosphorylate it in vitro. Several of the experiments demonstrate the same result therefore are repetitive.

The experiments performed with total cell lysates as kinase source are "crude" as several other kinases can phosphorylate VAR2A and the inhibitors can block them especially at high concentrations. Therefore, these may be excluded especially when kinase assays have been performed with purified recombinant hu/PfCk2 or with IP, which provide direct evidence of phosphorylation by the kinase although it is only in vitro.

3. The concentration of TBCA/DMAT used in experiments is in 10-50uM, which seems to be too high. What is the IC50 of these inhibitors against human and PfCK2.

4. Do Var2CSA and CK2- co-immunoprecipitate and co-localize in the parasite and/or IE?

5. Given that the ectodomain of Var2CA is phosphorylated by CK2, how does it come in contact with PfCK2? Does it get phosphorylated in the parasite before it is trafficked to the host ?

6. Figures 5-7: S1068 present in the CIDR is identified as the target phosphorylation site for hu/PfCK2a. There seems to be only a modest decrease in binding of S1068A mutant line to CSA. Statistical analysis of the data provided in Figure 6 should be performed.

7. Figure 7: The finding that trafficking of the S1068A mutant to the host RBC is impaired is interesting. However, more experiments need to establish that S1068A phosphorylation is critical for the export of Var2CSA to the host, which in turn is important for adhesion.

Here are a few suggestions:

a. The quality of IFA provided in Fig. 7c should be better and an additional marker for PVM and RBC surface may be included.

b. Figure 7D Western blot shows that S1068A is expressed at higher levels that WT Var2CSA. Was IE or only parasite lysate was used for this experiment? It may be good to fractionate RBC and parasite fractions and compare the WT and S108A mutant in these fractions.

c. BrefeldinA, which blocks protein transport, can be used to compare the trafficking of mutant and WT Var2CSA. In addition, co-localization with CK2 can be performed.

**Part III – Minor Issues: Editorial and Data Presentation Modifications**

Reviewer #1: 1. Similar to point one above, in the section entitled “Native VAR2CSA is phosphorylated by rHuCK2α”, the authors show that immunoprecipitated VAR2CSA can be phosphorylated by recombinant human CK2. However, they do not include a control to show that the immunoprecipitated protein is not contaminated with parasite CK2 (or any other kinase activity). This is particularly relevant considering that later in the manuscript they show that both the human and parasite kinases can be co-immunoprecipitated by anti-VAR2CSA antibodies. A simple control would be to perform the radiolabeling reaction with immunoprecipitated VAR2CSA but in the absence of recombinant CK2. The authors performed an equivalent control in the following section when they incubated immunoprecipitated VAR2CSA with an enzymatically dead recombinant PfCK2α and observed no phosphorylation, thus demonstrating that their immunoprecipitated protein does not contain significant contaminating kinase activity. It would be good to include a similar control in this experiment or at least mention the subsequent result.

2. Minor corrections: throughout the manuscript, the word “var” is sometimes italicized and sometimes not. In Figure 7, panel B, the word “VAR2CSA” is misspelled next to S1068A.

Reviewer #2: The quality of images and statistical analysis needs to be done as indicated in my comments

PLOS authors have the option to publish the peer review history of their article (what does this mean?). If published, this will include your full peer review and any attached files.

Reviewer #1: No

Reviewer #2: No
---

## [Decision Letter · Decision Letter 1]

23 Dec 2024

Dear Dr Gamain,

We are pleased to inform you that your manuscript 'Casein Kinases 2-dependent phosphorylation of the placental ligand VAR2CSA regulates Plasmodium falciparum-infected erythrocytes cytoadhesion' has been provisionally accepted for publication in PLOS Pathogens.

Best regards,

David A. Fidock, Ph.D.

Academic Editor

PLOS Pathogens

Tracey Lamb

Section Editor

PLOS Pathogens

Sumita Bhaduri-McIntosh

Editor-in-Chief

PLOS Pathogens

orcid.org/0000-0003-2946-9497

Michael Malim

Editor-in-Chief

PLOS Pathogens

orcid.org/0000-0002-7699-2064

Your revised manuscript has been carefully reviewed by one of the original reviewers, who concludes that it has satisfactorily addressed all the concerns and requests for additional experimentation. A member of the editorial board has also reviewed the resubmission and accompanying rebuttal and agrees. The only request for change was a reviewer comment to correct a gene ID to the standard PlasmoDB ID.

Reviewer Comments (if any, and for reference):

Reviewer's Responses to Questions

**Part I - Summary**

Reviewer #1: The manuscript provides molecular details regarding an important phosphorylation event that occurs on a protein called VAR2CSA. This protein is the key virulence factor expressed by parasites causing pregnancy associated malaria and is a high profile target for vaccine development. Understanding how the protein is processed provides important insights into possible intervention strategies.

**Part II – Major Issues: Key Experiments Required for Acceptance**

Reviewer #1: The initial review requested clarification for several experiments and the addition of some controls. These requests have been addressed by the authors. There are no additional major issues that must be addressed.

**Part III – Minor Issues: Editorial and Data Presentation Modifications**

Reviewer #1: In the pie chart shown in Figure 8A, the author denote var2csa as "PFL0030c". This is an old annotation that many readers will not be familiar with. I suggest that the authors change this to "Pf3D7_1200600".

PLOS authors have the option to publish the peer review history of their article (what does this mean?). If published, this will include your full peer review and any attached files.

Reviewer #1: No

---

## [Editor Report · Acceptance letter]

6 Jan 2025

Dear Dr Gamain,

We are delighted to inform you that your manuscript, "Casein Kinases 2-dependent phosphorylation of the placental ligand VAR2CSA regulates Plasmodium falciparum-infected erythrocytes cytoadhesion," has been formally accepted for publication in PLOS Pathogens.

Best regards,

Sumita Bhaduri-McIntosh

Editor-in-Chief

PLOS Pathogens

orcid.org/0000-0003-2946-9497

Michael Malim

Editor-in-Chief

PLOS Pathogens

orcid.org/0000-0002-7699-2064